



# Aircraft vertical profiles during summertime regional and Saharan dust scenarios over the north-western Mediterranean Basin: aerosol optical and physical properties

Jesús Yus-Díez[1,2], Marina Ealo[1,2], Marco Pandolfi[1], Noemí Perez[1], Gloria Titos[1,3], Griša Močnik[4,5], Xavier Querol[1], and Andrés Alastuey[1]

[1]Institute of Environmental Assessment and Water Research (IDAEA-CSIC), C/Jordi Girona 18-26, 08034, Barcelona, Spain
[2]Departament de Física Aplicada - Meteorologia, Universitat de Barcelona, C/Martí i Franquès, 1., 08028, Barcelona, Spain
[3]Andalusian Institute for Earth System Research (IISTA-CEAMA), University of Granada, Autonomous Government of Andalusia, Granada, Spain.
[4]Center for Atmospheric Research, University of Nova Gorica, Vipavska 11c, SI-5270 Ajdovščina, Slovenia.
[5]Department of Condensed Matter Physics, Jozef Stefan Institute, Jamova 39, SI-1000 Ljubljana, Slovenia

**Correspondence:** jesus.yus@idaea.csic.es

**Abstract.**

Accurate measurements of the horizontal and vertical distribution of atmospheric aerosol particle optical properties are key for a better understanding of their impact on the climate. Here we present the results of a measurement campaign based on instrumented flights over NE Spain. We measured vertical profiles of size segregated atmospheric particulate matter (PM) mass
concentrations and multi-wavelength scattering and absorption coefficients in the Western Mediterranean Basin (WMB). The campaign took place during typical summer conditions, characterized by the development of a vertical multi-layer structure, under both summer regional pollution episodes (REG) and Saharan dust events (SDE). REG patterns in the region form under high insolation and scarce precipitation in summer, favoring layering of highly-aged fine PM strata in the lower few km a.s.l. The REG scenario prevailed during the entire measurement campaign. Additionally, African dust outbreaks and plumes
from North African wildfires influenced the study area. The vertical profiles of climate relevant intensive optical parameters such as single scattering albedo (SSA), asymmetry parameter (g), scattering, absorption and SSA Angstrom exponents (SAE, AAE, SSAAE), and PM mas scattering and absorption cross sections (MSC and MAE) were derived from the measurements. Moreover, we compared the aircraft measurements with those performed at two GAW/ACTRIS surface measurement stations located in NE Spain, namely: Montseny (MSY; regional background) and Montsec d'Ares (MSA; remote site).
Airborne in-situ measurements and ceilometer ground-based remote measurements identified aerosol air masses at altitude up to more than 3.5 km a.s.l.. The vertical profiles of the optical properties markedly changed according to the prevailing atmospheric scenarios. During SDE the SAE was low along the profiles, reaching values <1.0 in the dust layers. Correspondingly, SSAAE was negative and AAE reached values up to 2.0-2.5, as a consequence of the UV absorption increased by the presence of the coarse dust particles. During REG, the SAE increased >2.0 and the asymmetry parameter g was rather low $(0.5 - 0.6)$
due to the prevalence of fine PM which were characterized by an AAE close to 1.0 suggesting a fossil fuel combustion origin. During REG, some of the layers featured larger AAE (>1.5), relatively low SSA at 525nm (<0.85) and high MSC (>9 $\mathrm{m^2\,g^{-1}}$)



and were associated to the influence of PM from wildfires. Overall, the SSA and MSC near the ground ranged around 0.85 and 3 $m^2\ g^{-1}$, respectively and increased at higher altitudes, reaching values above 0.95 and up to 9 $m^2\ g^{-1}$. The PM, MSC and MAE were on average larger during REG compared to SDE due to the larger scattering and absorption efficiency of fine PM compared with dust. The SSA and MSC had quite similar vertical profiles and often both increased with height indicating the progressive shift toward PM with larger scattering efficiency with altitude.

This study contributes to our understanding of regional aerosol vertical distribution and optical properties in the WMB and the results will be useful for improving future climate projections and remote sensing/satellite retrieval algorithms.

# 1   Introduction

Atmospheric aerosol particles play an important role in the Earth's radiative balance directly by scattering and absorbing solar radiation and indirectly by acting as cloud condensation nuclei (Myhre et al., 2013). Globally, the aerosol particles direct radiative effect is negative at the top of the atmosphere due to their net cooling on the Earth's atmosphere system (Myhre et al., 2013). This estimation is, however, affected by a large uncertainty which reflects the large spatial and temporal variability of the optical, physical and chemical properties of optically active atmospheric aerosol particles (Myhre et al., 2013). This variability is largely due to the wide variety of aerosol sources and sinks, their intermittent nature and spatial non-uniformity together with the chemical and physical transformations that aerosol particles undergo in the atmosphere (Myhre et al., 2013). These variables affect the scattering and absorption properties of atmospheric particles, and consequently their radiative properties. Thus, uncertainties remain on the effects that aerosol particles exert on climate from local to global scales, and a detailed characterization of their properties in the horizontal and vertical dimensions is needed (Andrews et al., 2011; Bond et al., 2013; Haywood et al., 1999; Collaud Coen et al., 2013). An accurate knowledge of their radiative forcing will improve both the climate reanalysis and forecasting models, needed in the current climate change context (Myhre et al., 2013). The in-situ surface distribution of atmospheric particles and their physical and chemical properties is determined through an array of networks across US and Europe and to a lesser extent in Asia and Africa (Laj et al., 2020), such as the Global Atmosphere Watch (GAW, World Meteorological Organization), the European Research Infrastructure for the observation of Aerosol, Clouds and Trace Gases (ACTRIS; www.actris.eu), the Interagency Monitoring of Protected Visual Environments (IMPROVE ; http://vista.cira.colostate.edu/Improve/), and the European Monitoring and Evaluation Programme (EMEP, www.Emep.int). These networks provide detailed optical, physical and chemical properties of atmospheric aerosol particles at the surface (Putaud et al., 2004, 2010; Andrews et al., 2011; Asmi et al., 2013; Collaud Coen et al., 2013; Cavalli et al., 2016; Zanatta et al., 2016; Pandolfi et al., 2018; Collaud Coen et al., 2020; Laj et al., 2020).

Given that the radiative forcing by aerosols is produced in the whole atmospheric column, the study of the vertical distribution of aerosol particles and their properties is of great importance (Sheridan et al., 2012; Samset et al., 2013). In fact, uncertainties on their vertical distribution and its relationship with surface emission sources are still a subject of intensive research. For example, Sanroma et al. (2010) suggested that, during cloud-free conditions, high-altitude aerosol particles are the major contributor to variations in solar radiation flux reaching the surface, at least at the high altitude sites they studied. Moreover,





the positive radiative forcing associated with strongly absorbing particles, such as black carbon (BC), is amplified when these particles are located above clouds (Zarzycki and Bond, 2010, and references therein) and the sign and magnitude of semi-direct BC effects depends on the BC relative location to the clouds and cloud type (Bond et al., 2013), with different clouds differently impacting the atmospheric heating rate of different aerosol types (Ferrero et al., 2020).

Ground-based remote sensing measurements of aerosol particles optical and physical properties are performed by a number of projects, such as AERONET (AErosol RObotic NETwork; https://aeronet.gsfc.nasa.gov), which is the international federation of ground-based sun and sky scanning radiometer, and the European Aerosol Research LIDAR Network (EARLINET/ACTRIS; www.earlinet.org), providing column integrated and vertically resolved optical properties, respectively. Satellite and sun/sky photometer measurements provide information on the aerosol properties of the entire atmospheric column, top-down and bottom-up, respectively. Polar-satellite observations provide a wide spatial coverage, yet are limited in time resolution to one or twice a day. On the other hand, laser based active instruments such as LIDARs (Light Detection and Ranging) or ceilometers provide reliable information of the vertically resolved optical properties of aerosols, such as backscattering and extinction. LIDARs measure at several wavelengths (e.g. Raman-elastic LIDARs; Ansmann et al., 1992), whereas ceilometers are usually limited to one wavelength.

The aforementioned remote sensing methods have thus limitations for directly measuring vertical resolved distribution of specific climate relevant optical parameters/variables such as SSA, asymmetry parameter (g) or absorption coefficients. Although not as frequent, airborne measurements can provide the vertical profiles and horizontal variability of these parameters, as well as of other aerosol related patterns. Instrumented flights are usually performed for specific campaigns allowing determining aerosol properties up to heights of a few km a.g.l in the free troposphere. For example, Esteve et al. (2012) performed a campaign over Bondville (Illinois) aiming at comparing AERONET remote sensing measurements with measurements of aerosol properties, such as scattering, absorption and other derived variables from instrumented flights. In Asia, Clarke et al. (2002) measured aerosol particles microphysical, optical and chemical properties over the Indian Ocean. Recently, Singh et al. (2019) measured particle size, number (N), spectral absorption, and meteorology variables in different pollution layers along a Himalayan valley in Nepal by means of instrumented flights. Vertically resolved measurement campaigns have been performed with tethered balloons, for example in the Po valley (Ferrero et al., 2014), in the Arctic (Ferrero et al., 2016), and with Unmanned Aerial Vehicles (UAVs), such as the ones over the Eastern Mediterranean by Pikridas et al. (2019). Aircraft aerosol measurements over Europe were performed for example during the EUCAARI-LONGREX campaign as shown by Highwood et al. (2012), who presented the results of a closure study between measured vertical aerosol particles scattering and absorption and Mie theory. The vertical profiles of total and accumulation mode N concentration, BC mass concentration, LIDAR measurements and the estimated radiative aerosol effect during EUCAARI-LONGREX were presented by Hamburger et al. (2010), Groß et al. (2013) and Esteve et al. (2016). In the context of the ChArMEx/ADRIMED instrumented flights measuring physical and chemical properties took place (Mallet et al., 2016; Denjean et al., 2016). These measurement campaigns have demonstrated the usefulness of instrumented flights providing aerosol properties which cannot be measured with conventional remote sensing techniques. These vertically resolved aerosol measurements are restricted to the timespan of the campaign as well as to the regional area where these take place. However, aircraft measurements might yield very relevant information for



detailed radiative forcing studies, and might be extremely useful for models aiming at simulating these effects over specific regions. Moreover, the results could be extrapolated to other periods when measurements are not available in the same area or to other areas with similar meteorological patterns and pollutant emissions.

Detailed studies of the vertical profiles of climate relevant aerosol properties are especially important in climate sensitive areas, such as the Mediterranean Basin. This is one of the regions in the world characterized by high loads of both primary and

secondary aerosol particles, especially in summer (Rodríguez et al., 2002; Dayan et al., 2017) from diverse emission sources. Anthropogenic emissions from road traffic, industry, agriculture, and maritime shipping, among others, strongly contribute to the air quality impairment in this region (Querol et al., 2009b; Amato et al., 2009; Pandolfi et al., 2014c). Moreover, the Mediterranean Basin is also highly influenced by natural sources, such as mineral dust from African deserts and smoke from forest fires (Bergametti et al., 1989; Querol et al., 1998; Rodrıguez et al., 2001; Lyamani et al., 2006; Mona et al., 2006; Koçak

et al., 2007; Kalivitis et al., 2007; Querol et al., 2009b; Schauer et al., 2016; Ealo et al., 2016; Querol et al., 2019, among others). In addition to this, the distribution of the different types of aerosol particles in the Mediterranean Basin is largely modulated by regional and large scale circulation (Millán et al., 1997; Gangoiti et al., 2001; Kallos et al., 2007). The complex orography surrounding the WMB sea and the high insolation, scarce precipitation and low winds during summertime give rise to high rates of accumulation and formation of secondary particles (Rodríguez et al., 2002). The NE of Spain provides a frame of study that

represents well the atmospheric summer conditions of the Western and Central Mediterranean, characterized by frequent and severe pollution episodes with high PM, UFP (ultra-fine particles) and $O_3$ formation (Pey et al., 2013; Pandolfi et al., 2014a; Querol et al., 2017, among others). Typical atmospheric dynamics coupled to local orography results in local/regional vertical recirculation with a consequent accumulation of pollutants (Millán et al., 1997). Vertical recirculation and ageing of pollutants is favored by weak gradient atmospheric conditions, scarce precipitation and continuous exposure to solar radiation driving

photochemical reactions (Rodríguez et al., 2002; Pérez et al., 2004). This recirculation is very relevant in mid-summer when the solar radiation increases and the high pressure periods last longer, so that the vertical recirculation reach further inland and create reservoir strata (loaded with pollutants) at 1-3 km a.s.l. (Gangoiti et al., 2001; Pérez et al., 2004). In fact, due to the geography and climatic summer meteorological patterns, formation of strong breezes from the coast through the valleys up to the Pyrenees and pre-Pyrenees range is frequent (Ripoll et al., 2014). Additionally, and especially in spring/summer, intense

Saharan dust outbreaks (Querol et al., 1998; Rodrıguez et al., 2001; Escudero et al., 2007; Querol et al., 2009b; Pey et al., 2013; Querol et al., 2019) and forest fires (Faustini et al., 2015) influence air quality over the WMB. The above atmospheric processes might coincide in space and time in the WMB and result in summer radiative forcing above this area being among the highest in the word (Lelieveld et al., 2002, and references therein). To the best of our knowledge, only few in-situ aircraft/balloon-borne measurement campaigns aiming at studying the optical properties of tropospheric aerosol particles were performed in the WMB

such as those performed within ChArMEx/ADRIMED campaign (the Chemistry-Aerosol Mediterranean Experiment/Aerosol Direct Radiative Impact on the regional climate in the MEDiterranean region; e.g. Denjean et al., 2016).

To better characterize the complex summer atmospheric scenarios of the WMB and to better understand their effect on the concentrations and optical properties of aerosols, we present here results from an aircraft measurement campaign performed within the PRISMA project in the north-western Mediterranean (PRISMA: Aerosol optical properties and radiative forcing in





the western Mediterranean as a function of chemical composition and sources). The main objective of the PRISMA project was to study the spatial and vertical distribution of the atmospheric aerosol particles with special interest in their physical and optical properties to assess the regional radiative forcing caused by the tropospheric aerosols in the WMB. To this end two measurement campaigns were performed in summers 2014 and 2015 combining aircraft measurements with remote and in-situ surface measurements performed at Montseny (MSY) and Montsec (MSA) stations. Herein we focus on the 2015 campaign, when vertical summer recirculation episodes, Saharan dust outbreaks, and plumes from wildfires simultaneously affected air quality over the WMB. Combination of the aircraft measurements with the available surface in-situ/remote measurements permitted to well characterize the summer recirculation coupled with Saharan dust outbreaks usually observed in the WMB in summer. The results presented here provide a more exhaustive characterization of the aerosol layers than the one that can be obtained by deploying in-situ surface and remote sensing techniques applied so far in the region.

In the following section (Sect. 2) we describe the methodology, with a deeper analysis of the study area, measurement stations and flight location as well as the main meteorological scenarios. In section 3 we will explain the main aerosol optical properties calculations. The main results obtained in this study are shown in section 4 for the different meteorological scenarios for both surface and the vertical measurements. Finally, we discuss the obtained results in the conclusion (Sect. 5).

## 2 Methodology

### 2.1 Area of study and meteorology

Airborne aerosol measurements were performed in an area of around 3500 $\mathrm{km}^2$ in the NW Mediterranean area of Catalonia, NE Spain (Fig. 1). The area is close to the Barcelona Metropolitan Area, where anthropogenic emissions, mostly from road traffic, industry, agriculture, and maritime shipping (Barcelona is one of the major ports in the Mediterranean) contribute to air quality impairment (Querol et al., 2001; Pey et al., 2013; Pérez et al., 2008; Amato et al., 2009; Reche et al., 2011; Amato et al., 2016, among others). Figure 1 shows the study area, the location of the Monsteny (MSY; 720 m a.s.l.) and Montsec (MSA; 1570 m a.s.l.) measurement stations and the location and ID codes (P1-P7) of the instrumented flights (Table 1).

This area is characterized by warm summers and temperate winters with irregular and rather scarce precipitation rates, especially in summer. The Azores high pressure system plays an important role in the synoptic meteorology of the Iberian Peninsula (IP). In winter, the Azores anticyclone is located at lower latitude and then the IP is more influenced by low-pressure systems and the associated fronts coming from the N Atlantic. However, in summer, the anticyclonic system intensifies and moves toward higher latitudes inducing very weak pressure gradient conditions over the IP. These atmospheric stagnant conditions, coupled to local orography and sea breezes circulation, result in local and regional atmospheric dynamics that favors the accumulation and ageing of pollutants (Millán et al., 1997; Gangoiti et al., 2001; Millán et al., 2002; Rodríguez et al., 2002; Pérez et al., 2004, among others). The high surface temperatures give rise to the formation of the Iberian Thermal Low, which induces the convergence of surface winds from the coastal areas injecting polluted air-masses into the middle troposphere ($3.5 - 5\,\mathrm{km}$ height). During the daytime the sea breeze layer (up to $800\,\mathrm{m}$ high) is channeled into the coastal and pre-coastal valleys up to $90\,\mathrm{km}$ inland, transporting pollutants from the coastal area (Gangoiti et al., 2001). On the mountain




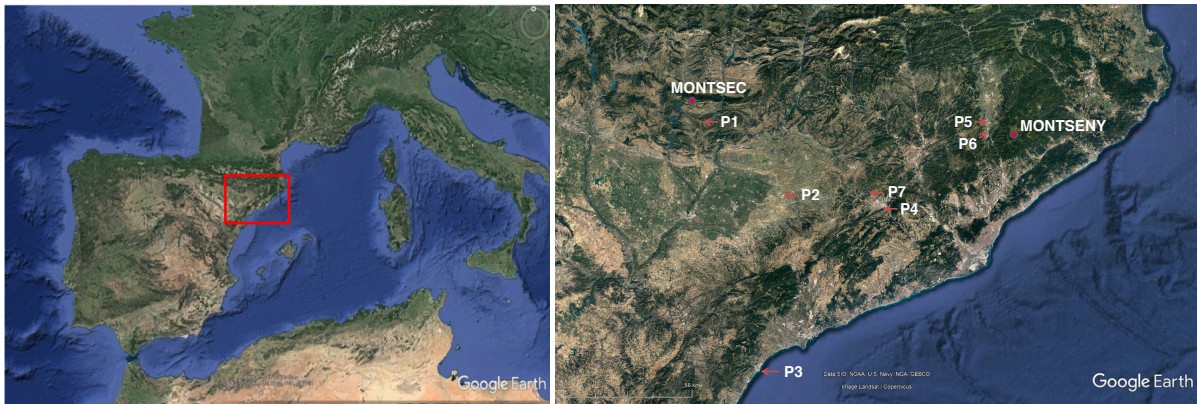

**Figure 1.** Western Mediterranean Basin and zoom over the NE of Spain with the MSA and MSY stations placed and the number ID of the instrumented flights (Table 1), located where measurements were taken. Satellite view from © Google Earth.

and valley slopes, up-slope winds inject pollutants at higher levels. Sea breezes and up-slope winds over the coastal and pre-coastal mountains transport coastal pollutants inland while a fraction of these pollutants is injected in their return flows aloft (2 − 3 km) forming high pollution loaded reservoir strata (Gangoiti et al., 2001). Pollutants are returned to the Mediterranean Sea owing to the prevalence of westerly winds above the mixing layer. Once in those upper layers, pollutants move back towards the sea and compensatory subsidence creates stratified reservoir layers of aged pollutants. Then, the next morning the lowermost layers are drawn inland again by the sea breeze (Millán et al., 2002; Gangoiti et al., 2001).

Saharan dust outbreaks affect the Iberian Peninsula mostly in spring/summer increasing the PM levels. The meteorological scenarios causing African dust transport over the study area are described in Rodrıguez et al. (2001) and Escudero et al. (2005). Four main scenarios are differentiated: North Africa High Located at Surface Level (NAH-S), Atlantic Depression (AD), North African Depression (NAD), North African High Located at Upper Levels (NAH-A). The WMB is mostly affected by the AD and NAD in spring and fall, by the NAH-A in summer and by the NAH-S from January to March. Another potential significant source of PM affecting air quality in the WMB are wildfires, which might persist for several days and emit large amounts of primary PM and precursors of secondary aerosols (Faustini et al., 2015). In addition to the complex atmospheric patterns, the scarce precipitation and the high exposure to solar radiation drive photochemical reactions resulting in high background levels of secondary aerosols and ozone (Rodríguez et al., 2002; Pérez et al., 2004; Querol et al., 2016, 2017).

### 2.1.1 Detailed synoptic and mesoscale meteorological scenarios during flights

Airborne measurements were conducted in 7th-8th and 14th-16th July 2015 to measure the optical properties of the vertical profile of the troposphere with the typical summer regional pollution episode and layering (Gangoiti et al., 2001). Hysplit backtrajectories in Fig. 2 (calculated for $7^{th}$ and $16^{th}$ July 2015) and the potential temperature vertical profile (Fig. S2) show that both periods were characterized by atmospheric stagnation (without major synoptic flows venting the WMB). Under these

**Figure 2.** Upper panel shows the backward trajectories at the MSA station, the dust concentration near ground as simulated by the Skyron Forecast Simulations (University of Athens) and the smoke surface concentration forecasted by the NAAPS Prediction System for 12:00 UTC of the 7th July 2015 (Flights P1 and P2). The lower panels shows the same information for 12:00 UTC of the 16th July 2015 (Flights P6 and P7).

summer atmospheric scenarios sea breezes and vertical re-circulation of air masses develop. However, during the first period an African dust outbreak affected the study area (Fig. 2), probably by high altitude transport from N Africa and subsequent impact on surface by convective circulations over the continent. Moreover, plumes from wildfires reached also the study area in the second period as the smoke forecast from NAAPS indicated (Navy Aerosol Analysis and Prediction System; http://www.nrlmry.navy.mil/aerosol; see Fig.2 ).

The geopotential height at $500\,\mathrm{hPa}$ and surface pressure from the ERA-INTERIM reanalysis model (https://www.ecmwf.int/en/forecasts/datasets/reanalysis-datasets/era-interim) show for the first day of the measurements a high pressure system





covering central Europe (Fig. 3 and Fig. S1). The 7th July (coincident with the first measurements), the Azores high pressure system and a N Europe low governed the synoptic flow. This situation persisted until 16th July, with an enhancement of the Iberian and North African Thermal Lows (Fig. 3).

Thus, during the campaign, the atmospheric scenario is the typical one favouring stagnation in the WMB, which leads to the vertical air masses re-circulations driven by local/regional processes (Gangoiti et al., 2001). Furthermore, the NE African high

pressure at high atmospheric levels (NAH-A; Escudero et al., 2005) favored the advection of Saharan dust at high altitude to the area of study from the 4th noon to the 8th afternoon July 2015. We will refer to this first period as SDE period, in reference to the Saharan Dust event that persisted during these days. On 14th and 16th July, smoke from wildfires over North Africa and the IP affected the study area. On this second period there was not mineral dust on 14th July, while on 16th July a light dust outbreak took place (Fig. 2). Due to the prevalence of summer regional pollution episodes with the absence of a dominant

strong Saharan dust event, we will refer to this second measurement period as REG.

## 2.2 Measurements and instrumentation

### 2.2.1 Ground supersites and measurements

Surface measurements were performed at Montseny (MSY, regional background) and Montsec (MSA, continental background) monitoring supersites (NE Spain). MSY ($41°46'46''$N, $02°21'29''$E, 720 m a.s.l.) is located in a densely forested area, 50 km

to the N–NE of the Barcelona, and 25 km from the Mediterranean coast. MSA ($42°03'05''$N, $00°43'46''$E, 1570 m a.s.l.) is located in a remote high-altitude emplacement in the southern side of the Pre-Pyreness at the Montsec d'Ares Mountain Range, at 140 km to the NW of Barcelona, and 140 km to the WNW of MSY (Fig. 1). Detailed descriptions of the measurement supersites and of the measurements performed can be found for example in Pérez et al. (2008); Pey et al. (2009); Pandolfi et al. (2011, 2014a, 2016) for MSY, and Pandolfi et al. (2014b); Ripoll et al. (2014); Ealo et al. (2016, 2018) for MSA. These

supersites are part of the Catalonian Air Quality Monitoring Network and are part of ACTRIS and GAW networks. Aerosol optical properties at the sites were measured following standard network protocols (WMO/GAW, 2016).

Similar instruments were used for in-situ surface characterization of physical, chemical and optical aerosol particle properties at both MSY and MSA. Aerosol particle total scattering ($\sigma_{sp}$) and hemispheric backscattering ($\sigma_{bsp}$) coefficients were measured every 5 min at three wavelengths (450, 525 and 635 nm) with a LED-based integrating nephelometer (Aurora 3000,

ECOTECH Pty, Ltd, Knoxfield, Australia). Calibration of the two nephelometers was performed four times per year using $CO_2$ as span gas, while zero adjusts were performed once per day using internally filtered particle-free air. The RH threshold was set by using a processor-controlled automatic heater inside the Aurora 3000 nephelometer to ensure a sampling RH of less than 40 % (GAW, 2016). $\sigma_{sp}$ coefficients were corrected for non-ideal illumination of the light source and for truncation of the sensing volumes following the procedure described in Müller et al. (2011b). Aerosol light absorption coefficients ($\sigma_{ap}$) at

seven different wavelengths (370, 470, 520, 590, 660, 880 and 950 nm) were obtained every 1 min at both stations by means of aethalometers (Magee scientific AE-33) (Drinovec et al., 2015). At both supersites a multi angle absorption photometer (MAAP, Model 5012, Thermo, Inc., USA Petzold and Schönlinner, 2004) was also used for obtaining the aerosol light ab-

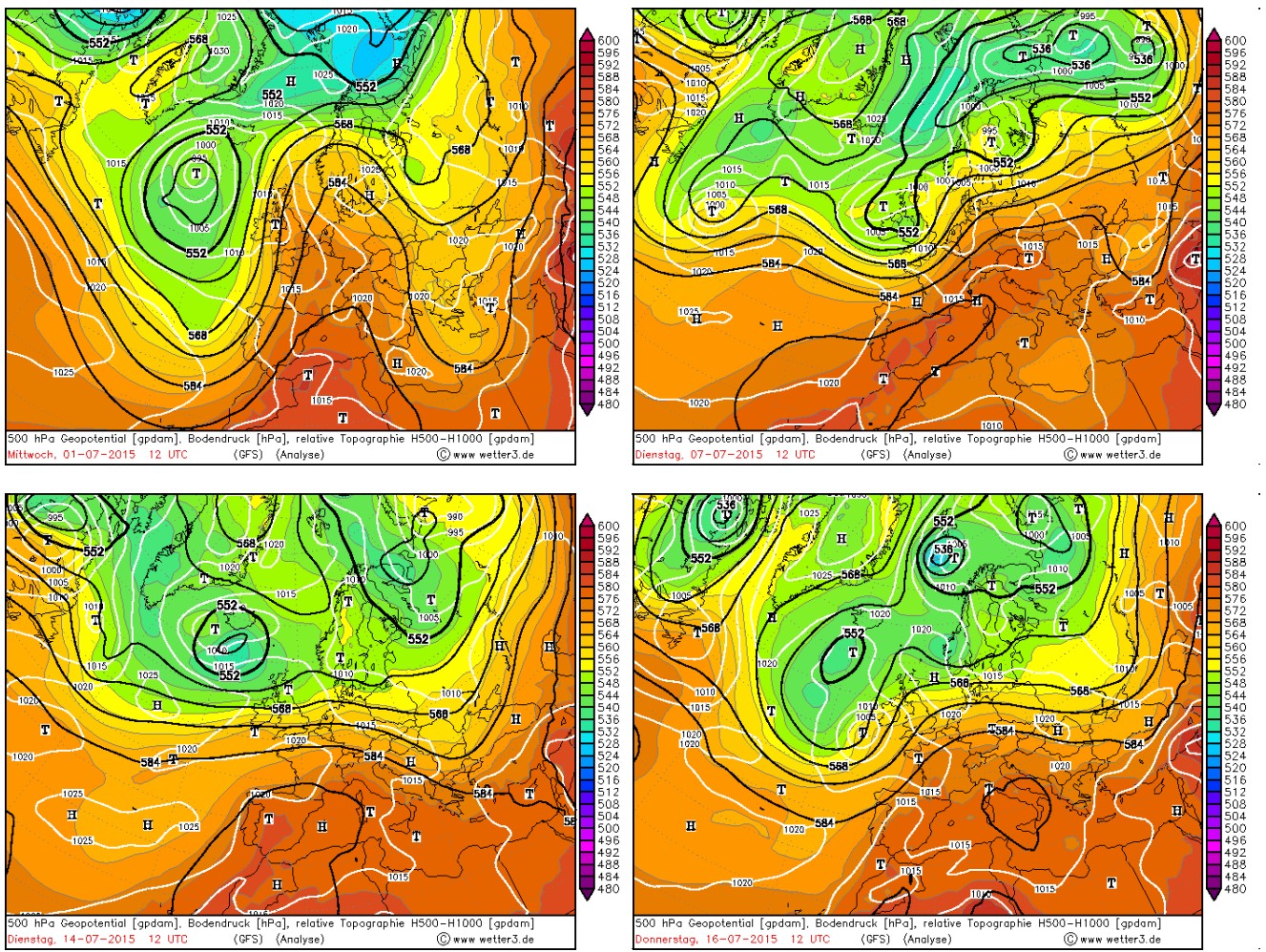

**Figure 3.** Synoptic meteorology with the $500\,\mathrm{hPa}$ Geopotential and surface pressure at 12:00 UTC for the 1st, 7th, 14th and 16th July of 2015 as modelled by the reanalysis model ERA-INTERIM, © www.wetter3.de.

sorption coefficient at a wavelength of 637 nm (Müller et al., 2011a). At both sites the instruments were connected to a $PM_{10}$ cut-off inlet. On-line surface in-situ optical measurements were reported to ambient temperature ($T$) and pressure ($P$) using

measurements from automatic and co-located weather stations.

PM mass concentrations were obtained with an optical particle counter (OPC) (Grimm 1108.8.80) connected to an individual inlet. Particle number volume concentrations were measured every $5\,\mathrm{min}$ in 15 size bins from $0.3$ to $20\,\mu\mathrm{m}$ and then converted to PM mass concentration by the instrument software. PM mass concentrations were corrected by comparison with standard gravimetric PM measurements (Alastuey et al., 2011) . An aerosol Chemical Speciation Monitor (ACSM, Aerodyne Research





Inc.) was used to measure real time non-refractory sub-micron aerosol species (organic matter, nitrate, sulphate, ammonium and chloride). Measurements with the ACSM at MSY were available from 4th to 12th July.

$PM_{10}$, $PM_{2.5}$ and $PM_1$ 24h samples were daily collected on $150$ mm quartz microfibre filters (Pallflex 2500 QAT-UP and Whatman QMH) using high-volume samplers (DIGITEL DH80 and/or MCV CAV-A/MSb at $30$ $m^3$ $h^{-1}$). The daily concentrations of major and trace elements and soluble ions (determined following the procedure by Querol et al. (2001)),

as well as those of organic (OC) and elemental (EC) carbon (by a thermal-optical carbon analyser, SUNSET, following the EUSAAR2 protocol (Cavalli et al., 2010)) were measured during the whole campaign. The ratio between the fine, $PM_1$, and coarse, $PM_{10}$, mode is obtained as the fraction between these two concentration measurements, $PM_{1/10}$. The chemical analyses allowed the determination of around 50 PM components, accounting for $75 - 85\%$ of the PM mass, the unexplained PM mass being mostly due to unaccounted moisture and non analyzed heteroatoms.

Active remote sensing measurements of attenuated backscatter ($\beta_{att}$) at $1064$ nm were performed at MSA with a Jenoptik CHM 15k Nimbus (G. Lufft Mess- und Regeltechnik GmbH, Germany) ceilometer. The instrument operated continuously with a temporal and spatial resolution of 1 min and 15 m, respectively. The maximum height of the signal is 15.36 km a.g.l.. Calibration based on Rayleigh calibration method was applied (Bucholtz, 1995; Wiegner et al., 2014). The overlap between the laser pulse and telescope field of views of the CHM 15k is greater than $60$ % at around $500$ m a.g.l. (Martucci et al., 2010;

Pandolfi et al., 2013). Further details of the ceilometer installed at MSA can be found in Titos et al. (2017).

Passive remote sensing measurements were obtained at MSA by means of a CE-318 sun/sky photometer (Cimel Electronique, France) included in AERONET.

### 2.2.2 Airborne measurements

Flights were performed with an aircraft Piper PA 34 Seneca (Fig.4) over NE Spain (Fig. 1) from around 0.5 up to 3.5 km a.s.l.,

with the same time duration (20-30 min), for both the upward and downward direction (Table 1). Both ascent and descent trajectories were performed in the same area when possible. For most of the cases, the up and down profiles were similar, showing therefore the representativeness of the measured profiles and a minimum interference of the aircraft emissions and turbulence.

The method used to perform the vertical profiles consisted in vertical ascensions following helical trajectories (Font et al.,

2008), thus allowing the measurement of the aerosol particles properties around a more constrained area. During the flights, an external flow controller (IONER PFC- 6020) was connected to the nephelometer in order to assure a constant sampling flow (5 $Lmin^{-1}$). Similarly to the in-situ surface measurements, the scattering measurements were performed at RH<40% (GAW, 2016). During the SDE period the vertical profiles of scattering were collected using the Kalman filter available in the AURORA3000 nephelometers, whereas during the REG period vertical profiles this filter was switched off and the raw $\sigma_{sp}$

and $\sigma_{Bsp}$ coefficients were collected. As shown later, the Kalman filter had the effect of smoothing the measured scattering and hemispheric backscattering coefficients.

Aerosol light-absorption coefficients measurements at seven different wavelengths (370 to 950 nm) were performed with the AE33 aethalometer (Magee scientific AE33, model AE33 AVIO). The AE33 AVIO is a modified prototype, based on the





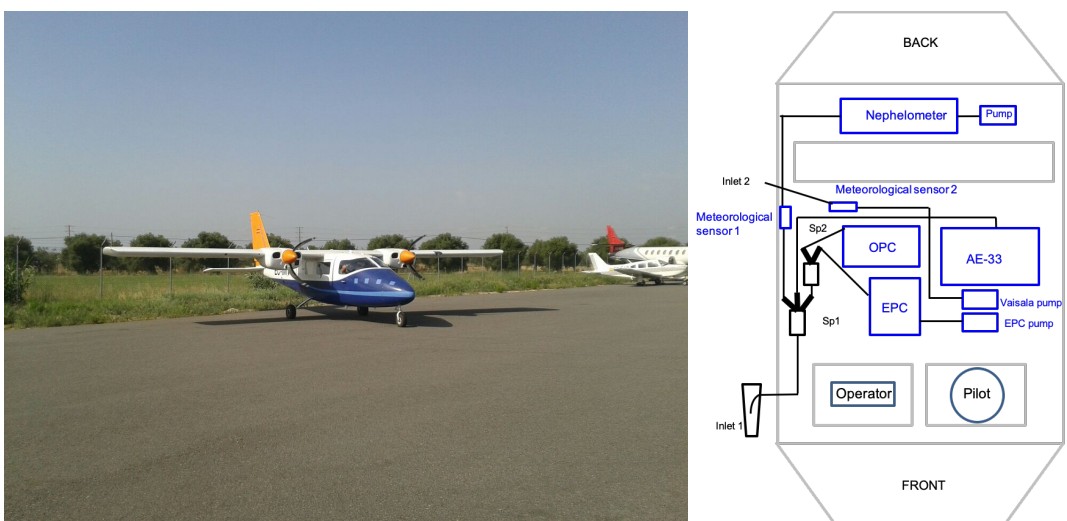

**Figure 4.** Piper PA 34 Seneca aircraft (left panel) and the schematic of the aerosol inlet and sampling instruments inside the plane (right panel). Aerosol sampling lines are shown in black and sampling instruments in blue.

AE33 aethalometer which was re-designed for aircraft measurements. The measurement and operational principle is identical to the commercial AE33 (Drinovec et al., 2015), but the AE33 AVIO model deploys a more powerful internal pump to ensure a constant flow ($4 \, \mathrm{L min^{-1}}$) with changing pressure. Aethalometer measurements were performed at 1 s resolution.

Particulate matter (PM) mass concentration was obtained using an aerosol optical particle counter (OPC) (GRIMM Spectrometer, model 1129-Sky-OPC). Particle number concentration was measured in 31 size bins at 6 s and at a flow rate of $1.2$ $\mathrm{L min^{-1}}$, for particles with size diameter from $0.25$ to $32 \, \mu\mathrm{m}$. The software instrument provides PM mass concentration, which is calculated assuming a constant particle density. Total particle number concentration was measured at 1 s by means of a water-base condensation particle counter (TSI WCPC, model 3788) at a flow rate of $0.6 \, \mathrm{L min^{-1}}$). The instrument provides size range down to $5 \, \mathrm{nm}$ and measured concentration range up to $2.5 \cdot 10^5 \, \# \, \mathrm{cm^{-3}}$.

Temperature (T) and relative humidity (RH) inlet sample air was measured at 15 s with a Rotronic HL-RC-B - Wireless logger. This is a passive sensor, so no flow regulator was needed to pass the air through the sensor. A Vaisala DRYCAP dewpoint transmitter, model DMT143 was used to measure dew point and water volume concentration of outdoor ambient air at 10 s via inlet 2. Ambient air was drawn out at 10 s and at a flow rate of $2 \, \mathrm{L min^{-1}}$. An external pump with a flow regulator was connected to the Vaisala sensor in order to maintain a constant flow. Pressure (P) data was obtained by means of a Kestrel 4000 anemometer. All the aerosol data was converted to T and P sample air conditions.

The inlet, manufactured by Aerosol d.o.o. (www.aerosol.si), was designed to be close to isokinetic thus minimizing the inlet losses. In order to determine aerosol sampling efficiency and particle transport losses we used the Particle Losses Calculator (PLC) software tool (von der Weiden et al., 2009). The results (not shown) indicated that the losses at the inlet are minimal





**Table 1.** Flight measurement location and date. Vertical profiles were performed up to heights of 3.5 km a.s.l..

| Flight | Date | Time (UTC) | Coordinates | Area |
|--------|------|-----------|-------------|------|
| P1 | 2015-07-07 | 08:55 - 09:16 | 41.96 N - 0.75 E | MSA |
| P2 | 2015-07-07 | 15:31 - 15:53 | 41.65 N - 1.14 E | Plain |
| P3 | 2015-07-08 | 08:11 - 08:31 | 41.01 N - 0.94 E | Coast |
| P4 | 2015-07-14 | 09:44 - 10:00 | 41.60 N - 1.64 E | Catalan Pre-coastal mountain range |
| P5 | 2015-07-14 | 10:56 - 11:12 | 41.85 N - 2.20 E | Catalan Pre-coastal mountain range |
| P6 | 2015-07-16 | 10:07 - 10:23 | 41.84 N - 2.24 E | Catalan Pre-coastal mountain range |
| P7 | 2015-07-16 | 10:55 - 11:15 | 41.61 N - 1.58 E | Catalan Pre-coastal mountain range |

for $PM_{2.5}$ and that the losses inside the sampling system were large for dust particles larger than around 4-5 $\mu m$. To further confirm the PLC results, we compared the $PM_x$ measurements at MSA station with the $PM_x$ aircraft measurements performed during the closest flight to MSA at approximately the same altitude of MSA station. This flight (P1; 7th July 2015; Fig. 1) was performed during a Saharan dust outbreak thus the presence of dust particles in the atmosphere was significant. Table S1 shows that differences were around 6% for $PM_1$ and 12% for $PM_{2.5}$ with the aircraft underestimating the measurements at MSA. These differences were low considering the spatial variation of the $PM_x$ concentrations between P1 flight and MSA station (approximately 10 km). However, the $PM_{10}$ measurements with the aircraft were around 47% lower compared to MSA thus confirming the PLC results. Therefore, the inlet losses for particles other than dust were minimal. In fact, Table S1 shows that for scattering and absorption measurements (which were performed in the $PM_{10}$ fraction at MSA) the differences were <9% and <16%, respectively because of the high scattering efficiency of fine particles compared to coarse particles (e.g. Malm and Hand, 2007) and because the absorbing fraction is mostly contained in the fine aerosol particle mode.

## 3  Determination of the intensive aerosol optical properties

The optical aerosol particle characterization was performed measuring the extensive optical properties ($\sigma_{ap}$, $\sigma_{sp}$, $\sigma_{Bsp}$) and calculating the intensive optical properties for both in-situ surface and aircraft borne measurements. The intensive optical properties (reported below) are mass independent properties which were determined starting from the multi-wavelength measurements of extensive properties. The intensive properties strongly depend on the physico-chemical aerosol properties. Hereafter there is brief description of the calculated intensive optical properties and the equations used to derive them:

a. The Scattering Ångström Exponent (SAE) depends on the physical properties of aerosols, and mostly on the particle size. SAE values lower than one are associated with the presence of coarse particles, whereas SAE>1.5 indicate the presence of fine particles (Seinfeld and Pandis, 1998; Schuster et al., 2006). The SAE was calculated as linear fit in a log-log space using the scattering measurements at the three wavelengths.





b. The asymmetry parameter (g) represents the probability of radiation being scattered in a given direction. The range of values is $[-1, 1]$ for backward ($180°$) to forward ($0°$) scattering, respectively. A value of around 0.7 is employed in the climate models (Ogren et al., 2006). The g was calculated from the Backscatter Fraction (BF)s using the semi-empirical formula provided by Andrews et al. (2006).

c. The Absorption Ångström Exponent (AAE) depends mostly on the chemical composition of sample particles. For pure BC, typical AAE values range from 0.9 to 1.1 (Kirchstetter et al., 2004; Petzold et al., 2013). When brown carbon (BrC) or mineral dust are sampled together with BC, then the AAE increases because both BrC and dust can efficiently absorb radiation in the ultraviolet and blue region of the spectrum compared to the near-IR (Kirchstetter et al., 2004; Chen and Bond, 2010). The AAE was calculated as linear estimation in the log-log space using the multi-wavelengths absorption measurements from AE33. For the vertical profiles reported here, the AAE was calculated using the seven AE33 wavelengths when all the seven absorption measurements were positive. For some profiles, the lower wavelengths (370 to 590, 660, or 880 nm) were used for the calculation of AAE.

d. The Single Scattering Albedo (SSA) reported here was calculated as the ratio between the scattering and the extinction coefficients at 525 nm. SSA indicates the potential of aerosols for cooling or warming the atmosphere.

e. The Single Scattering Albedo Ångström Exponent (SSAAE) was obtained as fit of the SSA calculated at the same Aethalometer wavelengths used to calculate the AAE. The SSA was obtained by extrapolating the total scattering at the Aethalometer wavelengths using the SAE. SSAAE is a good indicator for the the presence of coarse particles (e.g. dust) when values are <0 (Coen et al., 2004; Ealo et al., 2016).

f. The Mass Scattering Cross-section (MSC) is the ratio between the scattering and the PM concentration. It represents the scattering efficiency of the collected particles per unit of mass.

g. The Mass Absorption Efficiency (MAE) is the ratio between the light absorption and the PM concentration. It represents the absorption efficiency of the collected particles per unit of mass.

## 4 Results

### 4.1 MSA and MSY in situ measurements

The time evolution of $PM_1$,$PM_{10}$ concentrations,$PM_{1/10}$ ratios, BC, PM components (as measured by ACSM at MSY), $\sigma_{ap}$,$\sigma_{sp}$, SAE, AAE, g and SSA measured at MSY and MSA during the first three weeks of July 2015 is presented in Fig. 5 together with the concentrations of major species ($NO_3^-$, $SO_4^{2-}$, $NH_4^+$, EC, OM (with an OM/OC ratio of 2.1), mineral matter (MM; calculated as the sum of typical mineral oxides) and sea salt (SS; Na + Cl)) from offline analysis of 24h filters collected at MSY and MSA during the days of the instrumented flights. Table 2 shows the mean values of surface $PM_1$, $PM_{10}$, $PM_{1/10}$, $\sigma_{sp\ 525nm}$, $\sigma_{ap\ 637nm}$, SAE, AAE, SSA and g measured at MSY and MSA stations on 7th-8th and 14th-16th July





compared to the mean values typically measured during SDE and REG pollution episodes as reported by Pandolfi et al. (2014b) and Ealo et al. (2016).

As shown in Fig. 5 an accumulation of pollutants took place from 4th to 8th July, as evidenced by the gradual increase of concentrations of $SO_4^{2-}$, BC, OA, PM, total scattering and absorption coefficients. This accumulation was favored by the regional stagnation and vertical recirculation of the air masses. Moreover, a Saharan dust outbreak caused a progressive increase of $PM_{10}$ at both stations, as well as a simultaneous $PM_{1/10}$ ratio decrease (Fig. 5c,d). The dust event had a larger impact at MSA where the $PM_{10}$ levels increased sharply and were higher compared to MSY. As reported in Table 2 and in Fig. 5, starting from

6th July, the $PM_{10}$ concentrations were higher compared to average $PM_{10}$ usually measured during Saharan dust outbreaks at both stations. Then, the levels of pollutants, as well as the total scattering and absorption coefficients, decreased on 9th July due to the venting of the basin by an Atlantic North West (ANW) advection (not shown) that cleansed the northern area of the IP (Gangoiti et al., 2006). This decrease was again sharper at MSA than at MSY due to the location of MSA at the top of a high mountain (Fig. 5 c and d and Fig. S4). After this event, a new regional pollution episode occurred which favored again

the accumulation of pollutants until the end of the campaign. Furthermore, according to NAAPs modelling outputs, relatively high levels of smoke loads occurred in the air masses affecting the study area on 14th-16th July 2015 (Fig. 2).

The intensive optical properties, i.e. SAE (Fig. 5g), AAE (Fig. 5h), g (525 nm; Fig. 5i) and SSA (525 nm; Fig. 5j), showed values consistent with the accumulation process and with the occurrence of the coarse Saharan dust particles, as well as the effects of the wildfires occurring in the IP. The SAE progressively decreased at both stations from values >1.5-2.0 (indicating

the predominance of fine particles) on 4th July to values <1.0 (indicating the predominance of coarse particles) on 8th July 2015. The variability of g was less pronounced compared to the variability of the SAE, especially at MSA, likely because the lower sensitivity of the g parameter to coarse PM compared to SAE (Pandolfi et al., 2018). The AAE was considerably >1.0 at both sites, reaching values >1.5. The high AAE was due to enhanced UV absorption, which typically is observed when brown carbon (BrC) and/or mineral dust particles are present in the atmosphere (Kirchstetter et al., 2004; Fialho et al., 2005;

Sandradewi et al., 2008; Alastuey et al., 2011; Chen and Bond, 2010). Interestingly, the AAE measured in the whole column with the sun-photometer over MSA showed even larger values, up to 2.5, until 8th July evening. The cleansing effect of the ANW on 9th July caused a reduction of the PM loads and consequently of the measured scattering and absorption coefficients at both sites. During this ANW flow the SAE increased again to values of 1.5-2.0; whereas both the surface and column integrated AAE decreased due to the removal of dust by the ANW cleansing effect. As already mentioned, after the ANW

advection, another regional pollution episode developed until the end of the campaign.

In the following sections, we present a detailed description and interpretation of the evolution of pollutants and aerosol optical parameters measured at MSY and MSA during the SDE and REG period. The mean values of the considered pollutants and optical parameters during airborne measurements are reported in Table 2, together with the mean values typically register during SDE outbreaks and REG episodes.





**Saharan dust event period, SDE**


The attenuated backscatter from the ceilometer at MSA allowed identifying the occurrence of aerosol layers up to 5 km a.g.l. (i.e. 5.8 km a.s.l.) on this period (Fig. 6). The air masses featured significant dust concentrations which were probably further accumulated in the region by the regional vertical recirculation of air masses. Correspondingly, the PM$_{10}$ mass concentrations measured at MSY and MSA were higher compared to the PM$_{10}$ concentrations usually measured in dust outbreaks (Table 2). Mineral matter concentrations on these days (Fig 5b) reached 21 and 15 µg m$^{-3}$ at MSA and MSY, respectively. The OA occurred in very similar concentrations at both sites ( 7 µg m$^{-3}$). Daily concentrations of $SO_4^{2-}$ and EC from filters were higher in the regional background (MSY) compared to the continental one (MSA) due to the larger impact of regional sources such as industries and road traffic and the emissions from maritime shipping from the Mediterranean Sea and the port of Barcelona. As expected, $NO_3^-$ concentrations were rather low at both sites due to the thermal instability of nitrate in summer (Harrison and Pio, 1983; Querol et al., 2001, 2004, 2009a).



Fig. 5a shows that the evolution of $SO_4^{2-}$ concentration at MSY station, as recorded by the ACSM, was similar to that of BC, both reaching a maximum on 7th July evening, with two subsequent minor peaks at midnight and the evening of 8th July (Fig. 5). The peak just after midnight, at 00:00 UTC 8th July, was also reflected in the OA, $NO_3^-$ and $NH_4^+$ concentrations. At MSA, ACSM measurements were not available, however the measured absorption showed a progressive increase on 8th July with a relative maximum reached in the afternoon (Fig. 5d), when AAE at MSA reached values >2.0 (Fig.5). This sharp increase in BC was attributed to local biomass burning emissions (Fig. 2). During this first measurement period, the mean scattering and absorption coefficients at both sites (Table 3) were above the averaged values registered at MSA and MSY stations during dust and summer regional pollution events (Pandolfi et al., 2014b). Figure 5e also shows the high AOD from MSA sun/sky photometer during the SDE period.


Table 2 shows that during this period the intensive parameters SAE, AAE, SSA and g reached values similar to the typical coarse PM influenced ones recorded during dust outbreaks at both stations (Pandolfi et al., 2014b). As already pointed above, on 8th July afternoon a change due to local biomass burning emissions was observed in the optical aerosol properties at MSA compared to MSY. As a consequence, the SAE and AAE at MSA increased to values >1.5 and 2.0, respectively.


**Regional pollution episode, REG**


A REG episode developed during the second aircraft measurement period leading to the accumulation of atmospheric aerosols in the area. A wildfire outbreak from North Africa took also place (Fig. 2), as well as a light dust outbreak during the 16th July (Figs. 2 and 5). The REG episode resulted in the development of aerosol layers at high altitude, as also observed with the ceilometer at MSA (Figs. 6c,d). PM$_{10}$ concentrations were rather constant and quite similar at both MSA and MSY but on average lower than during the SDE period. Only on 16th July, PM$_{10}$ concentrations at MSA, due to the impact of the dust outbreak at the altitude of MSA station, where closer to a SDE scenario rather than a REG scenario (Table 2).This pattern was also reflected in the high PM$_{1/10}$ ratio measured from the 10th July with the exception of the 16th July afternoon at MSA when the PM$_{1/10}$ ratio was lower. The lower impact of the dust outbreak during these days compared to the first period was






**Table 2.** Mean values of in-situ surface $PM_1$, $PM_{10}$, $PM_{1/10}$, $\sigma_{sp\ 525nm}$, $\sigma_{ap\ 637nm}$, SAE, AAE, SSA and g measured at MSY and MSA stations during the SDE period of 7-8 of July, and the REG period of 14 and 16 of July compared to the mean values typically measured (denoted by "avg") during SDE and REG episodes as reported by Pandolfi et al. (2014a) and Ealo et al. (2016).

|  | MSA | | | | MSY | | | |
|---|---|---|---|---|---|---|---|---|
|  | $\mathbf{SDE_{7-8}}$ | $\mathbf{SDE_{avg}}$ | $\mathbf{REG_{7-8}}$ | $\mathbf{REG_{avg}}$ | $\mathbf{SDE_{7-8}}$ | $\mathbf{SDE_{avg}}$ | $\mathbf{REG_{7-8}}$ | $\mathbf{REG_{avg}}$ |
| $\mathbf{PM_1}$ **(µg m$^{-3}$)** | $9.7 \pm 5.6$ | $8.4 \pm 4.7$ | $7.1 \pm 3.0$ | $7.9 \pm 3.9$ | $14.1 \pm 4.6$ | $12.3 \pm 5.4$ | $14.0 \pm 4.6$ | $11.8 \pm 5.7$ |
| $\mathbf{PM_{10}}$ **(µg m$^{-3}$)** | $30.8 \pm 8.3$ | $21.0 \pm 17.0$ | $19.8 \pm 8.5$ | $12.6 \pm 7.0$ | $33.2 \pm 3.3$ | $25.4 \pm 17.0$ | $23.8 \pm 5.7$ | $15.6 \pm 8.0$ |
| $\mathbf{PM_{1/10}}$ **(%)** | $29.5 \pm 10.1$ | $39.6 \pm 1.1$ | $37.8 \pm 10.8$ | $46.2 \pm 2.1$ | $42.3 \pm 13.6$ | - | $60.0 \pm 16.1$ | - |
| $\mathbf{\sigma_{sp\ 525nm}}$ **(Mm$^{-1}$)** | $42.5 \pm 11.2$ | $53.6 \pm 30.4$ | $51.9 \pm 10.9$ | $47.0 \pm 21.9$ | $54.4 \pm 14.9$ | $60.2 \pm 29.8$ | $56.4 \pm 12.8$ | $50.6 \pm 24.6$ |
| $\mathbf{\sigma_{ap\ 637nm}}$ **(Mm$^{-1}$)** | $4.53 \pm 1.66$ | $2.84 \pm 0.03$ | $5.03 \pm 1.60$ | $2.50 \pm 0.03$ | $4.89 \pm 2.19$ | $4.04 \pm 2.27$ | $4.42 \pm 1.59$ | $4.26 \pm 2.35$ |
| **SAE** | $1.25 \pm 0.31$ | $1.33 \pm 0.61$ | $1.56 \pm 0.19$ | $1.81 \pm 0.30$ | $1.05 \pm 0.28$ | $1.50 \pm 0.53$ | $1.03 \pm 0.29$ | $1.92 \pm 0.31$ |
| **AAE** | $1.46 \pm 0.25$ | $1.41 \pm 0.25$ | $1.29 \pm 0.15$ | $1.27 \pm 0.24$ | $1.26 \pm 0.17$ | $1.27 \pm 0.24$ | $1.23 \pm 0.12$ | $1.24 \pm 0.19$ |
| **SSA** | $0.88 \pm 0.03$ | $0.93 \pm 0.06$ | $0.89 \pm 0.02$ | $0.92 \pm 0.05$ | $0.90 \pm 0.03$ | $0.91 \pm 0.03$ | $0.90 \pm 0.03$ | $0.88 \pm 0.04$ |
| **g** | $0.58 \pm 0.03$ | $0.60 \pm 0.05$ | $0.60 \pm 0.02$ | $0.57 \pm 0.05$ | $0.55 \pm 0.02$ | $0.60 \pm 0.04$ | $0.62 \pm 0.08$ | $0.58 \pm 0.25$ |

confirmed by the markedly lower MM from PM speciation analysis (Fig. 5b). Despite the higher MM at MSA, those of the remaining $PM_{10}$ components and BC were similar at both sites.

The scattering (Fig. 5e) and absorption coefficients (Fig. 5f) at both sites were above the average values (Table 2) presented by Pandolfi et al. (2014a) and Ealo et al. (2016) for REG episodes, confirming the accumulation of pollutants since the 10th July onward. AOD (Fig. 5e) during REG was rather low when compared to the SDE period due to the absence of dust, with the exception of 16th July, when AOD increased simultaneously with the light dust outbreak and probably the influence of wildfire pollution plumes.

The asymmetry parameter g showed similar values to those observed during the SDE period at both sites and did not change much during the dust event on 16th July because, as already observed, g is less sensitive to variations of coarse mode aerosols. Consequently, the relatively lower SAE on this day (especially visible at MSY) suggested variations in the larger end of the accumulation mode particles and above rather than in the lower end of the mode. The lower SAE at MSY compared to MSA during REG (and especially during 16 of July) was then probably due to a relative reduction of $PM_1$ at MSY compared to

the previous days whereas $PM_{10}$ kept similar values (Fig. 5) yet lower compared to the SDE period despite the light SDE event registered on 16 of July (Table 2). Moreover, the higher relevance of fine PM at MSA compared to the MSY regional background could also explain the observed difference. This could be due to a combination of factors including a higher exposure of MSA station to the fine PM from the wildfires, the lower impact of local/regional sources at MSA and the larger segregation of particles during transport toward MSA due to its remote location. During REG, AAE decreased compared to

the first period (especially at MSA), yet this was still >1.0, indicating a mixing of regional aerosols with BrC and mineral





dust. AAE of the whole column (sun/sky-photometer data) was also close or slightly >1.0. Finally, the SSA varied only little between REG and SDE at both sites.

## 4.2   In-situ vertical profiles

Here we focus on the vertical profiles obtained by means of airborne measurements during the two distinct SDE and REG
scenarios. Vertical profiles of optical extensive and intensive properties, as well as PM concentrations for upward and downward flights are reported in Table 1. Our objective is to analyse the vertical profile variation of these properties and their diurnal and spatial evolution and how these properties varied across layers in the troposphere. When possible, direct comparison with the surface data measured at MSA station was made.

As reported above, the comparison of airborne measurements at 1.5 km a.s.l. with in-situ surface measurements at MSA for
P1 reached a good agreement (Table S1), with a relative difference <10%, for $PM_1$ and $PM_{2.5}$ concentrations and the extensive and intensive optical properties. The large underestimation of the $PM_{2.5-10}$, missing -47 % in the airborne measurements, is the reason for excluding the $PM_{10}$ fraction from the vertical profiles. Consequently, we calculated the MSC and the MAE for $PM_{2.5}$.

**Saharan dust event, SDE**

Fig. 6a,b shows a clear atmospheric layering of pollutants from 0.8 to 5.0 km a.g.l. (i.e. 1.6 to 5.8 km a.s.l.) The magenta boxes in Fig. 6a,b highlights the time window when flights were performed. Figure 7 shows the three vertical profiles during the SDE period (Table 1, namely: P1 (Fig. 7a) and P2 (Fig. 7b), which were performed close to the MSA station (Fig. 1), and P3 (Fig. 7c) performed over the Catalan Pre-coastal Range (Fig. 1). We can observe from Fig. 7 that during the P1 and P3 flights the $PM_1$ and $PM_{2.5}$ concentrations were rather high in the upper atmosphere (4-12 and 7-17 $\mu g\ m^{-3}$ for $PM_1$ and $PM_{2.5}$,
respectively, above 2.5 km a.s.l.). At lower altitudes (<1,5 km a.s.l.) even higher PM concentrations were measured (10-16 and 16-22 $\mu g\ m^{-3}$ for $PM_1$ and $PM_{2.5}$, respectively) which were on the upper range of the typical surface concentrations reported in Ealo et al. (2016) and Pandolfi et al. (2014b) for SDE scenarios in the area. The vertical profiles of PM during P1 and P2 showed a progressive upwards reduction of PM concentrations, especially evident above 2.5 km a.s.l. in P2 (Figs. 6 and 7). The vertical profiles of scattering and absorption coefficients showed similar patterns to the ones described for PM with an upward
decrease. As already stated, the use of the Kalman filter in the nephelometer measurements of P1, P2 and P3 had a smoothing effect on the measured scattering coefficient, thus complicating a direct comparison with PM during these profiles.

The $PM_{1/2.5}$ ratio (Fig. 7) during P1 ranged from around 0.6 to 0.75 along the profile with a relative minimum between 2.5 and 3 km a.s.l. where the SAE reached its lowest values (<1). Conversely, a clear increase of the $PM_{1/2.5}$ ratio above 2.5 km a.s.l. was evident for P2, when the ratio reached 0.8. The values of both SSAAE and SAE were consistent with the $PM_{1/2.5}$
ratio found for the different layers in P1 and P2. The SAE during P1 and P2 ranged between 0.8 − 1.5 and the SSAAE kept negative values along both profiles with a pronounced minimum at 3.0 km a.s.l. layer during P1 highlighting a higher relative fraction of coarse particles attributed to the dust outbreak over the area of study. It should be noted that a direct comparison between the vertical profiles of $PM_{1/2.5}$ and SAE was difficult because of the Kalman filter which smoothed the calculated





intensive aerosol particles scattering properties and made difficult the comparison and especially for g, which involves both the

smoothed scattering and the hemispheric backscattering coefficients. For this reason, the g was not reported in Fig. 7. During P1 the AAE ranged between around 1.7 and 2.5 with the highest values measured above 2.5 km a.s.l. as a consequence of the presence of UV absorbing dust particles in the atmosphere. During P2, the AAE showed more constant values (around 2) with altitude mirroring the less variable SAE compared to P1. Both SSA and MSC showed increasing values with altitude during P1 and P2 ranging between 0.85-0.9 and 2-4 $m^2 g^{-1}$, respectively, whereas the MAE (at 525 nm) kept rather constant values

(around 0.2 $m^2 g^{-1}$) with altitude with slightly lower values above 2.0-2.5 km a.s.l..

Fig. 7c shows the vertical profile P3, which was consistent with P1 and P2 even if the calculated intensive optical properties suggested a reduced effect of dust particles during P3 compared to P1 and P2. PM concentrations during P3 were similar to P2 profile, although the $PM_{1/2.5}$ ratio was higher, indicating an increased importance of fine aerosol mode during P3 compared to P2. The scattering and absorption coefficients were similar to the values measured during P1 and P2, although the scattering

was higher during P3. The intensive optical properties of P3 indicated the predominance of coarse dust particles especially around 1.7 and 2.3 km a.s.l. where the SAE decreased up to values close to 1 and the SSAAE was negative. In this altitude range the AAE was on average higher compared to the rest of the profile and especially around 1.8 km a.s.l. where it reached a value of 2.5. As already noted, the observed AAE increase at these altitudes was due to the UV absorption from dust particles. Overall, during P3 the intensive properties such as SAE, AAE and SSA showed more variability with altitude compared to P1

and P2 as also reported by the ceilometer measurements which showed a more stratified atmosphere on 8 of July (P3) compared to 7 of July (P1 and P2; Figs. 6a,b).

During P1, P2 and P3, the AAE varied considerably with altitude and ranged between 1.5-2.5 depending on the flight and altitude. These values were higher than those measured in-situ at MSA and MSY stations, as shown in shown in Fig. 5 and where higher compared to the typical AAE values measured during SDE at these stations (Table 2). The lower AAE measured at

MSA and MSY stations compared to the AAE in the free troposphere measured during P1, P2 and P3 was due to the increased relative importance of BC particles close to the ground and within the PBL that reduced the AAE at the two measurement stations compared to the values obtained with the flights in the free troposphere. The UV absorption properties of African dust observed during P1, P2 and P3, and the corresponding increase of AAE, were driven by the presence of iron oxides, such as hematite and geothite, which efficiently absorb radiation at shorter wavelengths (e.g. Alfaro et al., 2004). The obtained $PM_{2.5}$

MAE values (around 0.2 -0.25 $m^2 g^{-1}$) observed in the dust layers during P1, P2 and P3 were consistent with the MAE values, around 0.24 $m^2 g^{-1}$, recently reported for dust particles by (Drinovec et al., 2020).

For the P1, P2 and P3 profiles both SSA and MSC progressively increased with altitude (Fig. 7). SSA in P1, P2 and P3 was rather similar and ranged from 0.83 to 0.93 from 1.5 to 3.5 km a.s.l.. Higher SSA at higher altitudes was due to different factors including the progressive reduction of the relative importance of BC particles with altitude and the presence of more efficient

scatterers at higher altitudes, together with the presence of dust, as evidenced by the higher MSC and lower MAE. The MSC during the three flights ranged from around 2 $m^2 g^{-1}$ at 1.5 km a.s.l. to 3-4$m^2 g^{-1}$ at 3.5 km a.s.l.. Pandolfi et al. (2014b) reported a MSC of $PM_{2.5}$ at MSA station of $3.3 \pm 1.9$, which was consistent with what observed with airborne measurements at the MSA altitude (around 1.5 km a.s.l.). Pandolfi et al. (2014b) also showed that on average the highest MSC values at MSA





were observed in summer, mostly because of aerosol ageing and the formation of efficient scatterers, such as secondary $SO_4^{2-}$

particles and OA during summer regional pollution episodes. In the area under study OA is expected to be mostly secondary organic aerosols (SOA) (e.g. Querol et al., 2017). Recently, Obiso et al. (2017) assessed the sensitivity of simulated MSC of MM, OA and $SO_4^{2-}$ by perturbing particle microphysical properties, such as size distribution, refractive index, mass density and shape. Perturbations were performed by changing by $\pm 20\%$ the widely used OPAC (Optical Properties of Aerosols and Clouds; Hess et al., 1998) reference values. (Obiso et al., 2017) compared the simulated MSC with experimental MSC of PM

and different PM components determined at MSY station. High MSC, up to 4-5 $m^2\ g^{-1}$ were simulated by Obiso et al. (2017) for OAs and $SO_4^{2-}$, and both were more sensitive to perturbations in refractive index than to variations in the size distribution. Lower MSC was reported by the same study for MM (0.5-2 $m^2\ g^{-1}$), and it was found to be more sensitive to perturbations of the size distribution of dust particles compared to other particle properties. Ealo et al. (2018) calculated the MSC for different pollutant sources retrieved by applying the positive matrix factorization (PMF) model on $PM_{10}$ chemically speciated data

collected at MSA station. Ealo et al. (2018) found that the MSC changed considerably depending on the sources considered, being the highest for secondary $SO_4^{2-}$ and the lowest for sources such as MM and SS (both mostly associated with coarse particles) and for anthropogenic sources, such as traffic or industrial (also emitting less efficient scattering particles such as BC). Thus, the observed increase of MSC with altitude could be due to the relative increase of $SO_4^{2-}$ and OA particles, causing important changes in the refractive indexes and size distributions with altitude. These changes could be related to the ageing of

the particles which is expected to be larger for particles at higher altitudes.

**Regional pollution episode, REG**

Fig. 8 shows the vertical profiles P4, P5, P6 and P7 over the MSY station and the Catalan Pre-coastal Range (Fig. 1 and Table 1). The vertical profile of the attenuated backscatter from the ceilometer (Fig. 6) showed layering of aerosol with altitude for this period. As detailed below, the observed aerosol layers probably originated mostly from the vertical recirculation of air

masses during summer regional pollution episodes (Gangoiti et al., 2001). Moreover, a PM outbreak from wildfire plumes during this period and a dust outbreak on 16th July were also detected (Fig. 2).

In the P4 profiles, the PM concentrations, scattering and absorption coefficients reached higher values below the 1.5. km a.s.l. layer (i.e. within the PBL where most of the pollutants generated at ground were trapped), and decreased with altitude. The high $PM_{1/2.5}$ ratio (> 0.95) for altitudes >1.5. km a.s.l. demonstrated the prevalence of fine particles in the troposphere

compared to the PBL. Accordingly, along the profile, SAE was high (1.5-2.7), indicating the predominance of fine particles, and SSAAE was positive, indicating the absence of dust particles. Correspondingly, the asymmetry parameter g decreased with altitude above 1.5. km a.s.l. down to 0.5 confirming the progressive shift toward smaller particles with altitude and an increased effect of particles in the lower end of the accumulation mode. Notable was the decrease of SAE (below 1.5) and SSAAE (close to 0), and the corresponding increase of g (up to 0.8), at around 3.3 km a.s.l., during P4 indicating the presence of coarse

particles at this height. AAE kept values between 0.8 and 1.2 along the whole P4 profile, suggesting the lack of strong UV absorbers (such as BrC or mineral dust), and the prevalence of absorbing BC particles from fossil fuel combustion (e.g. Bond et al., 2004; Zotter et al., 2017). Furthermore, the MAE showed higher variability compared to P1, P2 and P3 profiles reaching





values up to 0.3 suggesting a relative increasing importance of BC particles in the troposphere compared to P1, P2 and P3. As already observed during the SDE period, both SSA and PM$_{2.5}$ MSC increased with altitude, especially above 2 km a.s.l.,

suggesting the presence of efficient scatterers at these altitudes. Note that, as observed during the first period of measurements, the MSC at 1.5 km a.s.l. (2-3 m$^2$ g$^{-1}$) was consistent with the typical MSC recorded at MSA (Pandolfi et al., 2014b). At higher altitudes the MSC reached values close to 5 m$^2$ g$^{-1}$ with SSA around 0.95.

Profile P5 (Fig. 8b) was obtained one hour after P4 and the distance between the two profiles was about 55 km (Table 1 and Fig. 1). The main differences between P4 and P5 were the consistently higher total scattering and absorption coefficients and

PM concentrations at all heights during P5 compared to P4, and the presence of two layers at 2 and 3 km. a.s.l. along P5. As in P4, PM concentrations in P5 were dominated by the fine mode (a PM$_{1/2.5}$ >0.9), as confirmed by a high SAE (>1.75), a positive SSAAE along the profile and rather low g values which decreased to 0.5 at 2 km a.s.l.

The absorption coefficient during P5 was dominated by BC particles from fossil fuel combustion up to 2 km a.s.l with AAE values around 1.0 or lower below 2 km a.s.l. and with SSA and MSC reaching the lowest values around 0.7 and 1, respectively,

at 2 km a.s.l. indicating the predominance of absorbing BC particles at this altitude. From this layer upwards, SAE remained rather constant (1.8 – 2-2), whereas g, SSA, AAE and MSC progressively increased with altitude. Again, the increase of g with altitude could be indicative of variations of particle size with altitude and a progressive shift toward particles in the lower end of the accumulation mode. Particles in this size range are efficient scatterers at the nephelometer wavelengths thus explaining the larger MSC. These results together with the progressive increase of AAE and positive SSAAE suggested the presence of

fine BrC particles in the free troposphere probably from wildfire smoke as also shown by modelling outputs in in Fig. 2. The presence of a clear separated layer at around 2.5-3 km a.s.l. was also shown by ceilometer measurements (Fig. 6c).

The P6 profile Fig. 8c was obtained close to P5 (4 km distant) on 16th July (two days after P4 and P5; Table 1). Scattering and absorption coefficients and PM concentrations were higher compared to P5 as a consequence of the progressive accumulation of particles in the area due to the development of the regional pollution episode. This hypothesis was also supported by the

higher aerosol load and layering in the troposphere compared to the 14th July according the ceilometer measurements (Fig. 6). Both scattering coefficient and PM concentrations decreased above the PBL (estimated to be at 1.75 km a.s.l.), with a subsequent increase at higher altitudes where these variables kept rather constant values close to those observed within the PBL. Conversely, the absorption coefficient did not increase above the PBL and, consequently, the SSA increased (up to 0.93-0.97) in the free troposphere compared to the SSA values obtained within the PBL. The MSC showed a similar profile to SSA

reaching values around 3.5-5.0 m$^2$ g$^{-1}$, whereas the MAE decreased with altitude above the PBL. The relative abundance of fine particles during P6 was also supported by SAE values around 1.75 and low g values around 0.5-0.6 and a positive SSAAE along the whole profile. The AAE ranged between 0.8 and 1.2 indicating that the absorption was dominated by BC particles.

The profile P7 was obtained around 30 min. after P6 and at a distance of around 60 km from P6, and about 5 km from P4. Scattering and absorption coefficients and PM concentrations from P7 were rather similar to P6. The scattering and absorption

coefficients in the green and PM$_1$ concentrations within the PBL (below 1.75 km a.s.l.) reached 50 Mm$^{-1}$, 5 Mm$^{-1}$ and 20 $\mu$g m$^{-3}$, respectively. As for P6, scattering, absorption and PM decreased above the PBL and then increased again in the free troposphere, with a relative maximum at 2.0 - 2.5 km a.s.l., followed by a further increase with height, up to 3.5 km a.s.l.





Again, $PM_{1/2.5}$ (0.95), SAE (1.75-2.0), a positive SSAAE and g (0.5-0.6) indicated the prevalence of fine particles in the free troposphere above the PBL during P7. As observed during the previous flights, SSA and MSC had very similar profiles. In P7, two peaks were observed at 1.8 and 2.5 km a.s.l., with a marked minimum at 2.2 km a.s.l. The similarity of SSA and MSC profiles is attributed to the relative high proportion of non-absorbing particles compared to the absorbing particles (high SSA), and the high scattering efficiency of the sampled particles. These two SSA and MSC peaks were also simultaneous with a slight increase of AAE indicating an aerosol mixture with BrC particles. These AAE-SSA-MSC parallel profiles were persistent for the whole REG period. The 1.8 km a.s.l. peak was coincident with a reduction of scattering, absorption and PM concentrations just above the PBL, and could be associated to the vertical diffusion along the day of a layer which, as shown in Fig. 6d, at 00:00 UTC 16th July was at 2 km a.s.l.. The lower SSA, MSC and AAE at 2.2 km a.s.l. were coincident with a relative increase of scattering, absorption and PM concentrations, and indicated the presence of a layer with a larger relative proportion of BC particles. From there upwards, the subsequent observed layer can be associated to the one at 3 km a.s.l. at midnight. If we consider also the ground MSA data (Fig. 5) and those from P6, the first of these two layers can be associated to a reservoir strata (a recirculating layer according to Gangoiti et al. (2001)) above the PBL, which, as day progressed and the convective dynamics intensified, was then integrated into the PBL. Conversely, the largest layer can be related to light mineral dust outbreak and smoke from wildfire plumes affecting the PBL (and MSA station) later in the day. This was supported by the high albedo, with SSA >0.9 across the whole profile and increasing with height. The AAE was close to 1.0 for the whole profile, with peaks up to 1.5 where the PM concentration was at the lowest concentrations. This could be caused by a higher relative concentration of more brownish, aged and not very fine black particles, which was related to reservoir strata, produced by the vertical recirculation of air masses, with lower SAE and higher g, SSA and MSC.

## 5 Conclusions

We reported on the results of an aircraft measurement campaign aiming at studying the vertical profiles of physical properties (size segregated PM mass concentration and multi-wavelength scattering and absorption coefficients) of atmospheric particles in the Western Mediterranean Basin (WMB). Seven vertical profiles following helical trajectories were obtained in 7th-16th July 2015 over an area of 3500 $km^2$ in the North-east Spain. The measurements campaign was carried out under typical summer regional pollution scenarios, with vertical recirculation of air masses that cause interlayering of polluted layers in the first few km a.s.l (Gangoiti et al., 2001). These aged aerosol rich layers are driven by complex atmospheric dynamics, driven by the summer atmospheric stagnation, high insolation, low precipitation and intricate orography surrounding the WMB. We measured during two regional pollution episodes, and these already complex scenarios were also affected by two African dust outbreaks and long-range transported plumes of wildfires. The summer regional pollution episodes finished by venting of the polluted air masses by synoptic W and NW flows, a frequent occurrence in the WMB (Gangoiti et al., 2001, 2006).

We measured vertical profiles of $PM_1$ and $PM_{10}$ concentrations, and a set of climate relevant aerosol optical parameters – single scattering albedo (SSA), asymmetry parameter (g), scattering and absorption Angstrom exponents (SAE and AAE) and PM mass scattering cross section (MSC). These intensive optical properties depend on the microphysical and chemical



properties of atmospheric aerosol particles rather on their mass and are key input parameters in climate models. Simultaneous in-situ surface measurements performed at two monitoring supersites in the region( Montseny, MSY, regional background, 720 m a.s.l.; Montsec, MSA; remote; 1570 m a.s.l.) were also used to better characterize the aerosol particles sampled during the instrumented flights.

Ceilometer profiles indicated the occurrence of aerosol layers over the region up to more than 5 km a.s.l., and ground measurements indicated that mean $PM_1$ and $PM_{10}$ mass concentrations and the scattering and absorption coefficients were similar or even higher than the typical ones obtained in the region during summer regional pollution episodes and African dust outbreaks.

During all flights, PM concentrations, scattering and absorption were high up to around 3.5 km a.s.l. (maximum altitude
reached by the aircraft), detecting the regional layering of aerosol-rich strata at these altitudes. The first measurement period was affected by an African air mass outbreak with dust particles mixed with vertically recirculated regional aerosols; whereas during the second period, the typical summer regional pollution episode and the advected wildfires smoke plumes dominated over the area under study. The measured optical properties were distinctively affected by these two different scenarios. During the dust outbreaks, SAE was rather low along the profiles, <1.0 in the high-dust loaded layers, where AAE increased up to 2.0-
2.5, as a consequence of the high UV absorption was enhanced by the presence of coarser dust particles. During the regional pollution dominated scenario, SAE reached higher values (>2) and g asymmetry parameter was rather low (0.5 – 0.6); in this case due to the prevalence of fine primary and secondary particles, mostly from regional anthropogenic emissions and the favourable conditions for secondary aerosol formation (high insolation, relative stagnation, high biogenic emissions, among others). Furthermore, the vertical variation of AAE was not large, with values close to 1.0 (pointing to a high proportion of
fossil fuel combustion aerosols), with the exception of few layers with increased AAE, probably associated with the influence of wildfires related aerosols. MSC was on average higher during the regional pollution episodes compared to the dust outbreaks due to the higher scattering efficiency of fine particles with diameter closer to the wavelength of the sampling visible light compared to coarse particles. Overall, MSC increased with altitude ( 2 $m^2$ $g^{-1}$ near the surface up to 4-5 $m^2$ $g^{-1}$ in the upper levels) as the distance to the top of the polluted PBL (where particle absorption was on average higher), and the presence of
more efficient scatterers at higher altitudes. A previous modelling study on MSC constrained with experimental measurements performed at the MSY stations suggested that the observed high MSC at higher altitudes might be due to the predominance of fine organic (mostly secondary) and inorganic (mostly sulphate in summer) aerosols. The MSC and SSA vertical profiles were rather similar during the flights with the SSA also increasing with altitude and with a vertical variability that depended on the composition of the observed layers. Typical SSA values along the profiles ranged between 0.85 (near surface) and 0.95 (higher
altitudes), with a minimum of <0.85 in polluted layers where smoke from wildfires was probably present.

The results presented here provide a unique input for climate models aiming at studying the regional radiative and climate effects of atmospheric aerosol particles in the WMB. We presented robust vertically resolved measurements of intensive aerosol particles optical properties in the WMB troposphere where the well-known particle layering driven by the regional pollution episodes accompanied by vertical recirculation takes place especially in summer. We have shown that the distribution of aerosol



particles and their optical properties vary vertically along the layers formed during several days under the typical high-PM summer regional pollution regime, as well as by the strength of the advection of aerosol particles such as dust and smoke.

*Data availability.* The data used in this study are available from the corresponding authors upon request.

*Author contributions.* AA designed the research experiment; MP, NP, ME, GT and AA performed the instrumented flights as well as main-tained the in-situ measurement stations. GM designed the inlet and helped with the instrument settings on the aircraft. ME and GT extracted
the data from the instruments as well as prepared the data-sets. AA, MP and XQ played a crucial role in the processes of shaping the manuscript structure as well as helping with the data analysis. JYD developed the data process, the analysis of the results, and summarized and expressed them in this article. All authors provided advice regarding the manuscript structure and content as well as contributed to the writing of the final manuscript.

*Competing interests.* The authors declare that they have no conflict of interest. GM was, at the time of the aircraft campaign, but not during
the data analysis or manuscript writing, employed by the manufacturer of the Aethalometer AE33.

*Acknowledgements.* Measurements at Spanish sites (Montseny, Montsec and Barcelona) were supported by the Spanish Ministry of Econ-omy, Industry and Competitiveness and FEDER funds under the project HOUSE (CGL2016-78594-R), by the Generalitat de Catalunya (AGAUR 2014 SGR33, AGAUR 2017 SGR41 and the DGQA) and the European Commission via ACTRIS2 (project 654109). Marco Pan-dolfi is funded by a Ramón y Cajal Fellowship (RYC-2013-14036) awarded by the Spanish Ministry of Economy and Competitiveness.
Gloria Titos is funded by MINECO under postdoctoral program Juan de la Cierva (FJCI-2014-20819 and IJCI-2016-29838). We would also like to acknowledge Yolanda Sola for providing access to the sun/sky photometer data at MSA station and to Aerosol d.o.o. for lending the AVIO aethalometer. We acknowledge support of the publication fee by the CSIC Open Access Publication Support Initiative through its Unit of Information Resources for Research (URICI).



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

**Figure 5.** Ground-based physical and chemical measurements at MSA and MSY stations: a) the evolution of sub-micrometric chemical composition as measured by the ACSM and the AE33, b) the off-line chemical analysis of 24-hr quartz filters from both MSA and MSY stations for the instrumented flights days, c) and d) the temporal evolution of $PM_1$, $PM_{10}$ and the $PM_{1/10}$ ratio at MSY and MSA, respectively. The evolution during the measurement campaign of the the scattering coefficient by the integrating nephelometer and the aerosol optical depth (AOD) by the ceilometer and sun/sky photometer, and absorption by the MAAP is shown in e) and f), the evolution of the intensive optical properties such as the scattering Angstrom exponent (SAE), from both integrating nephelometer and columnar ceilometer and sun/sky derived scattering, the absorption Angstrom exponent (AAE), the asymmetry parameter (g), and the single scattering albedo (SSA at 525 nm) is shown in g), h), i) and j), respectively. The shadowed sections highlight the vertical profile measurement periods with the aircraft.



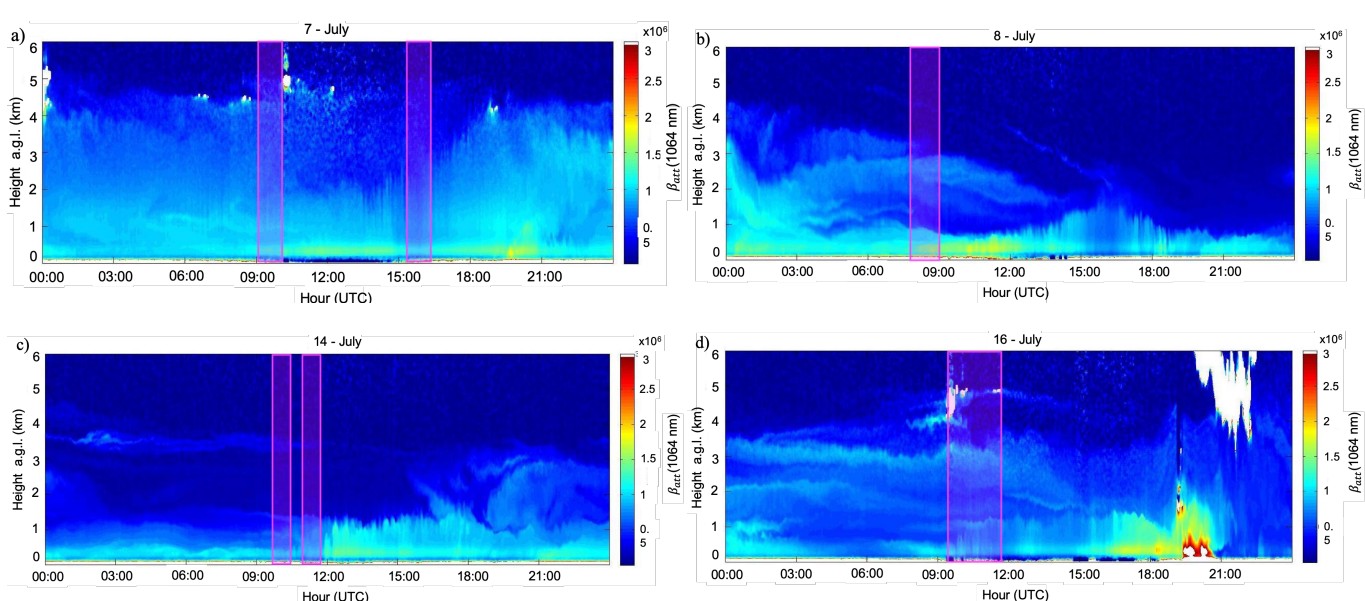

**Figure 6.** Time evolution $\beta_{a}tt$ vertical profiles as retrieved by the ceilometer deployed at MSA station during the July 2015 measurement campaign for the a) 7th, b) the 8th, c) the 14th and the d) 16th of July. The shadowed boxes highlight the measurements during the flights periods.





**Figure 7.** Vertical profiles from the extensive and intensive optical properties during the SDE period: scattering and absorption coefficients, $\sigma_{sp}$ and $\sigma_{ap}$, respectively, for several wavelengths, $PM_1$, $PM_{2.5}$ concentration and $PM_{1/2.5}$ ratio, single scattering albedo (SSA), scattering Angstrom exponent (SAE), absorption Angstrom exponent (AAE), single scattering albedo Angstrom exponent (SSAAE), mass scattering cross-section (MSC) and mass absorption efficiency (MAE) from left to right, for the aircraft borne measurements for the a) P1 and b) P2 and c) P3 vertical profile.





**Figure 8.** Vertical profiles from the extensive and intensive optical properties, as in Fig. 7 plus the asymmetry parameter (g), for the aircraft borne measurements during the REG period for the a) P4, b) P5, c) P6 and d) P7 vertical profiles around the Catalan Pre-coastal mountain range close to the MSY station.