# Peer review of "Aircraft vertical profiles during summertime regional and Saharan dust scenarios over the north-western Mediterranean Basin: aerosol optical and physical properties"

_Atmospheric Chemistry and Physics, 2020_

## Referee Comment (RC1) · Anonymous Referee #1 · 7 Oct 2020

This paper discusses aircraft measurements of aerosol optical and physical properties in the Western Mediterranean Basin. The measurements focused on two summertime aerosol events, one being an intrusion of Saharan dust into the study area and the other largely a regional pollution episode. The aircraft measurements took place near two instrumented ground stations so that comparisons during near-collocated sampling periods could be performed to tie the surface in situ measurements to the lower column airborne measurements.

General Comments: I thought the paper was well written and the data well presented.

[Figure]

This is a strong group scientifically with a breadth of knowledge and experience in both surface and airborne aerosol measurements. They know how to make the measurements correctly and have extracted a lot of valuable intensive aerosol property data from the fundamental measurements. I gave 'Excellent' grades to the broad categories of scientific significance, scientific quality, and presentation quality. Under scientific significance, 'excellent' may be a little high but I believe 'good' is too low. Probably it should be rated 'very good'. There is not really 'a substantial contribution to scientific progress...' since most of the methods are not really novel, but the data are new and there have been few aircraft programs to date to investigate these aerosol events in this region. Models need real data to initialize and challenge them, and to tell them when they are getting it right or wrong. This paper makes a good contribution in this regard. There are a number of smaller things in the manuscript that the authors need to clean up. I recommend that it be considered for publication in ACP after attention to my comments below and the other public and reviewers(s) comments.

Specific Comments: I have some comments about specific items in the manuscript. These are listed below.

Abstract, and also in the body of the paper: The authors use '...mas (sic) scattering and absorption cross sections (MSC and MAE). These are both 'mass (scat/abs) cross sections' and are also known as 'mass (scat/abs) efficiencies'. They are exactly the same except that one of these references scattering and the other references absorption. Why not use MSE and MAE (my preference)? Or else use MSC and MAC? I do not see any reason to possibly cause some confusion among people as to what these acronyms mean.

Lines 41-49: Should also include the NOAA Federated Aerosol Network (NFAN, Andrews et al., BAMS, 2019) in this listing, since MSA and MSY are in the NFAN.

Line 64: Replace 'Polar-satellite observations...' with 'Polar-orbiting satellite observations...'.

Lines 71-73: Not all airborne campaigns are of short-duration as your statement suggests. The study reported in Sheridan et al. (2012) had flights 2-3 times per week for 3.25 years!

Line 81: EUCAARI-LONGREX. Please define all acronyms and abbreviations the first time they are used in the manuscript.

Line 85: ChArMEx/ADRIMED. Same comment. Please define all acronyms and abbreviations the first time they are used in the manuscript.

Lines 96-97: '. . .strongly contribute to the air quality impairment. . .'. Replace with '. . .negatively affect the air quality. . .'.

Lines 143-144: '. . .contribute to air quality impairment. . .'. Same comment as before. Replace with '. . .negatively affect the air quality. . .'.

Line 146: '. . .and the location and ID codes (P1-P7) of the instrumented flights (Table 1).' These are 7 single points on the map. What do they represent? Center of the flight area? Center of profile location? The aircraft covers some horizontal area with its sampling. Possibly better to show shaded areas (or larger dots) for each flight.

Line 175: '. . .(calculated for 7th and 16th July 2015). . .'. Why was a trajectory at the start of your first period and at the end of the second period chosen? Were the trajectories consistent throughout each period? If so, why not calculate trajectories in the middle of each period?

Fig. 2 caption: 'The lower panels shows the same information. . .'. It should be noted that the scale on the trajectory map is different in the two panels.

Line 224: Replace 'aerosol' with 'Aerosol'.

Lines 241-242: Include this sentence in the previous paragraph. Should not have a one-sentence paragraph.

Section 2.2.2., first paragraph: Somewhere in this section the sampling inlet needs to

be discussed. How far away from the fuselage is the aerosol inlet? How far forward (assuming it is forward) of the leading edge of the wing is the inlet? How far away is the propeller? What is the total flow into this inlet? What is the typical true air speed of the airplane? There is a photograph in Fig. 4 of the airplane which is too small to show any of this information, and dimensions and inlet orientation relative to the wing are not provided in the drawing.

Lines 245-248: Were the data from both ascending and descending profiles used in this paper. It is only stated that for most of the cases, the data from the up and down profiles were similar.

Line 249: Replace 'consisted in...' with 'consisted of...'.

Line 249: This statement suggests that only the ascending profiles were used. Please clarify.

Line 249: 'Helical ascensions' implies that the aircraft was continuously turning. Was the aerosol inlet oriented to account for turning so that aerosols came straight into the inlet? If not, the aspiration efficiency of an inlet not precisely aligned with the flight streamlines should be checked. Was it? Another point... unless you have a moveable inlet that can adjust in flight, it is better to stick with one type of profile (ascending, descending, or level flight segments) so that the inlet orientation relative to the streamlines is constant. The pitch of a small aircraft can be different by as much as 10 degrees during ascending and descending profiles, and can also vary with fuel load and air speed. This also causes misalignment of the inlet with the air streamlines going past the inlet and could affect the aspiration efficiency of particles into the inlet. Do you have any calculations or fluid dynamics modeling that suggest this is not important (because you are not discussing it). The fact that aerosol data are similar in 'most' ascending and descending profiles provides circumstantial evidence that this is not a major concern but testing/modeling the effect would provide additional strong evidence.

Lines 251-252: '...in order to assure a constant sampling flow (5 Lmin−1).' Was the

flow controller a mass flow controller or a volume flow controller?

Lines 257-258: 'Aerosol light-absorption coefficients measurements at seven different wavelengths (370 to 950 nm) were performed with the AE33 aethalometer...'. How were the light absorption coefficients derived for the AE-33 aethalometer (i.e., which correction method was used to derive the aerosol light absorption coefficients from the raw aethalometer data)?

Line 274: 'The inlet, manufactured by Aerosol d.o.o. (www.aerosol.si), was designed to be close to isokinetic...'. At what flow rate was it designed to be close to isokinetic?

Lines 282-283: 'These differences were low considering the spatial variation of the PMx concentrations between P1 flight and MSA station (approximately 10 km).' Up until this point you have not presented any data showing horizontal inhomogeneity of aerosols in the region.

Lines 307-309: 'For the vertical profiles reported here, the AAE was calculated using the seven AE33 wavelengths when all the seven absorption measurements were positive. For some profiles, the lower wavelengths (370 to 590, 660, or 880 nm) were used for the calculation of AAE.' Why was the method not kept consistent for all profiles?

Lines 313-314: 'The SSA was obtained by extrapolating the total scattering at the Aethalometer wavelengths using the SAE.' Supposed to be 'SSAAE' instead of SSA? How confident are you in extrapolating the scattering, which is measured in the visible range, into the near UV and near IR?

Lines 323-324: '...presented in Fig. 5...' Some text in Fig. 5 barely readable. Font should be a little larger if possible.

Lines 387-388: 'The REG episode resulted in the development of aerosol layers at high altitude, as also observed with the ceilometer at MSA (Figs. 6c,d). How do you know that the features in these retrievals are not from light or subvisible cirrus rather than aerosols?

Lines 395-397: 'The scattering (Fig. 5e) and absorption coefficients 395 (Fig. 5f) at both sites were above the average values (Table 2) presented by Pandolfi et al. (2014a) and Ealo et al. (2016) for REG episodes, confirming the accumulation of pollutants since the 10th July onward.' I would not phrase this statement in this way. The fact that the scattering and absorption values here were higher than those reported in other studies from different years DOES NOT confirm the accumulation of pollutants over this period in your study. These are separate and unrelated events. Pollutants were indeed accumulating during this study, but it is not because they were higher than previously-measured average values. This point should be rephrased in the manuscript.

Line 411: '. . .the SSA varied only little. . .' Replace 'little' with 'slightly'.

Lines 426-427: 'Figure 7 shows the three vertical profiles during the SDE period. . .'. Same question as earlier. . . are these data from ascent profiles, descent profiles, an average of the two profiles, or what? If you are averaging two profiles, you can be averaging out real features in both profiles. Also, please state what the shaded envelopes represent for the intensive profile properties in Figs. 7 and 8.

Lines 445-447: Certainly there was dust, but can you rule out BrC or OA also contributing to the higher AAE's?

Line 458: 'As already noted, the observed AAE increase at these altitudes was due to the UV absorption from dust particles.' Still not sure how you have ruled out BrC/OA co-existing with the dust. It happens frequently with Asian dust where the pollution aerosols are transported along with the dust because the air mass travels over both dust and pollution sources and picks up both types of aerosols.
* * *

---

## Referee Comment (RC2) · Anonymous Referee #2 · 12 Oct 2020

Summary: The manuscript summarizes findings from an aircraft campaign over the western Mediterranean Basin in conjunction with in-situ aerosol measurements at two monitoring sites- MSY and MSA. The analysis explores two main episodes during the campaign, designated as (1) regional pollution episodes and (2) Saharan dust events. The paper synthesizes measured aerosol optical properties- including extensive and intensive properties- from the campaign and highlights the differences in meteorology and aerosol populations during the two types of episodes. While no new methods are presented, it is clear that care has been taken with the data and the analysis. The

paper is very well written and the analysis is clear.

General Comments:

-Excellent background section!

-My main qualm with the paper is that Figure 7 and 8 are quite difficult to read. It is as if the ratio of the figures is off; they should be slightly wider. I realize the challenges associated with presenting this much information at once- and I see the logic behind how these figures are organized. If the individual plots can be widened slightly or maybe if the legends could be moved to inside the plots to allow them to get bigger, that might help the reader. But the paper could be published with the figures as is if necessary.

Technical Comments:

Line 46: Perhaps add a reference to the NOAA Federated Aerosol Network here as well?

Line 65: Suggest changing to '…one or two a day' or '…once or twice a day'

Line 111-112 : Grammatically, 'reach' should be 'reaches' and 'create' should be 'creates'

Line 175: Suggest changing 'hysplit' to 'HYSPLIT' since their website capitalizes it

Figure 2: It would be nice to add the units of measurement to the figure descriptions in the caption, as they are difficult to read on the plots

Line 216: Please specify if aethalometer data were further corrected or if the manufacturer 2-spot correction was the correction used

Line 216: Were any corrections applied to the MAAP data?

Line 265: Extra ')' on this line can be deleted

Lines 307-309: It sounds like AAE was calculated using different wavelength pairs for

different legs. What is the effect of this on your results? Why not use a wavelength pair available in all flight leg data records so you have consistency?

Line 323: Add a space after 'concentrations,' before 'PM1/10'

Line 324: Add a space after 'sigma[ap]'

Line 358: It seems like 'typically register' should be past tense 'typically registered'

Line 386: should say 'also took place'

Line 536: How was the PBL estimated?

Lines 559-560: 'associated to' should be 'associated with'

Line 595: remove 'was'

Line 601: 'wildfire-related' instead of 'wildfires related'

Line 603-605: This sentence is difficult to read; consider rewording.

———————————————

---

## Author Response (AR1)

This paper discusses aircraft measurements of aerosol optical and physical properties in the Western Mediterranean Basin. The measurements focused on two summertime aerosol events, one being an intrusion of Saharan dust into the study area and the other largely a regional pollution episode. The aircraft measurements took place near two instrumented ground stations so that comparisons during near-collocated sampling periods could be performed to tie the surface in situ measurements to the lower column airborne measurements.

General Comments: I thought the paper was well written and the data well presented.

This is a strong group scientifically with a breadth of knowledge and experience in both surface and airborne aerosol measurements. They know how to make the measurements correctly and have extracted a lot of valuable intensive aerosol property data from the fundamental measurements. I gave 'Excellent' grades to the broad categories of scientific significance, scientific quality, and presentation quality. Under scientific significance, 'excellent' may be a little high but I believe 'good' is too low. Probably it should be rated 'very good'. There is not really 'a substantial contribution to scientific progress. . .' since most of the methods are not really novel, but the data are new and there have been few aircraft programs to date to investigate these aerosol events in this region. Models need real data to initialize and challenge them, and to tell them when they are getting it right or wrong. This paper makes a good contribution in this regard. There are a number of smaller things in the manuscript that the authors need to clean up. I recommend that it be considered for publication in ACP after attention to my comments below and the other public and reviewers(s) comments.

Specific Comments: I have some comments about specific items in the manuscript. These are listed below.

Abstract, and also in the body of the paper: The authors use '. . .mas (sic) scattering and absorption cross sections (MSC and MAE). These are both 'mass (scat/abs) cross sections' and are also known as 'mass (scat/abs) efficiencies'. They are exactly the same except that one of these references scattering and the other references absorption. Why not use MSE and MAE (my preference)? Or else use MSC and MAC? I do not see any reason to possibly cause some confusion among people as to what these acronyms mean.

Lines 41-49: Should also include the NOAA Federated Aerosol Network (NFAN, Andrews et al., BAMS, 2019) in this listing, since MSA and MSY are in the NFAN.

Line 64: Replace 'Polar-satellite observations. . .' with 'Polar-orbiting satellite observations. . .'.

Lines 71-73: Not all airborne campaigns are of short-duration as your statement suggests. The study reported in Sheridan et al. (2012) had flights 2-3 times per week for 3.25 years!

Line 81: EUCAARI-LONGREX. Please define all acronyms and abbreviations the first time they are used in the manuscript.

Line 85: ChArMEx/ADRIMED. Same comment. Please define all acronyms and abbreviations the first time they are used in the manuscript.

Lines 96-97: '...strongly contribute to the air quality impairment...'. Replace with '...negatively affect the air quality...'.

Lines 143-144: '...contribute to air quality impairment...'. Same comment as before. Replace with '...negatively affect the air quality...'.

Line 146: '...and the location and ID codes (P1-P7) of the instrumented flights (Table 1).' These are 7 single points on the map. What do they represent? Center of the flight area? Center of profile location? The aircraft covers some horizontal area with its sampling. Possibly better to show shaded areas (or larger dots) for each flight.

Line 175: '...(calculated for 7th and 16th July 2015)...'. Why was a trajectory at the start of your first period and at the end of the second period chosen? Were the trajectories consistent throughout each period? If so, why not calculate trajectories in the middle of each period?

Fig. 2 caption: 'The lower panels shows the same information...'. It should be noted that the scale on the trajectory map is different in the two panels.

Line 224: Replace 'aerosol' with 'Aerosol'.

Lines 241-242: Include this sentence in the previous paragraph. Should not have a one-sentence paragraph.

Section 2.2.2., first paragraph: Somewhere in this section the sampling inlet needs to

be discussed. How far away from the fuselage is the aerosol inlet? How far forward (assuming it is forward) of the leading edge of the wing is the inlet? How far away is the propeller? What is the total flow into this inlet? What is the typical true air speed of the airplane? There is a photograph in Fig. 4 of the airplane which is too small to show any of this information, and dimensions and inlet orientation relative to the wing are not provided in the drawing.

Lines 245-248: Were the data from both ascending and descending profiles used in this paper. It is only stated that for most of the cases, the data from the up and down profiles were similar.

Line 249: Replace 'consisted in...' with 'consisted of...'.

Line 249: This statement suggests that only the ascending profiles were used. Please clarify.

Line 249: 'Helical ascensions' implies that the aircraft was continuously turning. Was the aerosol inlet oriented to account for turning so that aerosols came straight into the inlet? If not, the aspiration efficiency of an inlet not precisely aligned with the flight streamlines should be checked. Was it? Another point... unless you have a moveable inlet that can adjust in flight, it is better to stick with one type of profile (ascending, descending, or level flight segments) so that the inlet orientation relative to the streamlines is constant. The pitch of a small aircraft can be different by as much as 10 degrees during ascending and descending profiles, and can also vary with fuel load and air speed. This also causes misalignment of the inlet with the air streamlines going past the inlet and could affect the aspiration efficiency of particles into the inlet. Do you have any calculations or fluid dynamics modeling that suggest this is not important (because you are not discussing it). The fact that aerosol data are similar in 'most' ascending and descending profiles provides circumstantial evidence that this is not a major concern but testing/modeling the effect would provide additional strong evidence.

Lines 251-252: '...in order to assure a constant sampling flow (5 Lmin−1).' Was the

flow controller a mass flow controller or a volume flow controller?

Lines 257-258: 'Aerosol light-absorption coefficients measurements at seven different wavelengths (370 to 950 nm) were performed with the AE33 aethalometer...'. How were the light absorption coefficients derived for the AE-33 aethalometer (i.e., which correction method was used to derive the aerosol light absorption coefficients from the raw aethalometer data)?

Line 274: 'The inlet, manufactured by Aerosol d.o.o. (www.aerosol.si), was designed to be close to isokinetic...'. At what flow rate was it designed to be close to isokinetic?

Lines 282-283: 'These differences were low considering the spatial variation of the PMx concentrations between P1 flight and MSA station (approximately 10 km).' Up until this point you have not presented any data showing horizontal inhomogeneity of aerosols in the region.

Lines 307-309: 'For the vertical profiles reported here, the AAE was calculated using the seven AE33 wavelengths when all the seven absorption measurements were positive. For some profiles, the lower wavelengths (370 to 590, 660, or 880 nm) were used for the calculation of AAE.' Why was the method not kept consistent for all profiles?

Lines 313-314: 'The SSA was obtained by extrapolating the total scattering at the Aethalometer wavelengths using the SAE.' Supposed to be 'SSAAE' instead of SSA? How confident are you in extrapolating the scattering, which is measured in the visible range, into the near UV and near IR?

Lines 323-324: '...presented in Fig. 5...' Some text in Fig. 5 barely readable. Font should be a little larger if possible.

Lines 387-388: 'The REG episode resulted in the development of aerosol layers at high altitude, as also observed with the ceilometer at MSA (Figs. 6c,d). How do you know that the features in these retrievals are not from light or subvisible cirrus rather than aerosols?

Lines 395-397: 'The scattering (Fig. 5e) and absorption coefficients 395 (Fig. 5f) at both sites were above the average values (Table 2) presented by Pandolfi et al. (2014a) and Ealo et al. (2016) for REG episodes, confirming the accumulation of pollutants since the 10th July onward.' I would not phrase this statement in this way. The fact that the scattering and absorption values here were higher than those reported in other studies from different years DOES NOT confirm the accumulation of pollutants over this period in your study. These are separate and unrelated events. Pollutants were indeed accumulating during this study, but it is not because they were higher than previously-measured average values. This point should be rephrased in the manuscript.

Line 411: '...the SSA varied only little...' Replace 'little' with 'slightly'.

Lines 426-427: 'Figure 7 shows the three vertical profiles during the SDE period...'. Same question as earlier... are these data from ascent profiles, descent profiles, an average of the two profiles, or what? If you are averaging two profiles, you can be averaging out real features in both profiles. Also, please state what the shaded envelopes represent for the intensive profile properties in Figs. 7 and 8.

Lines 445-447: Certainly there was dust, but can you rule out BrC or OA also contributing to the higher AAE's?

Line 458: 'As already noted, the observed AAE increase at these altitudes was due to the UV absorption from dust particles.' Still not sure how you have ruled out BrC/OA co-existing with the dust. It happens frequently with Asian dust where the pollution aerosols are transported along with the dust because the air mass travels over both dust and pollution sources and picks up both types of aerosols.

On behalf of all the authors, we thank the Reviewer for the positive comments and careful review, which helped improve the manuscript.

Hereafter we reply to the Reviewer's comments. Any minor comment as typo or writing corrections will be directly corrected in the manuscript.

**General comments**

- We thank the Reviewer for the positive comment regarding the work done by our research group and regarding the scientific significance of the present manuscript.

**Specific comments**

**Abstract, and also in the body of the paper: The authors use '. . .mas (sic) scattering and absorption cross sections (MSC and MAE). These are both 'mass (scat/abs) cross sections' and are also known as 'mass (scat/abs) efficiencies'. They are exactly the same except that one of these references scattering and the other references absorption. Why not use MSE and MAE (my preference)? Or else use MSC and MAC? I do not see any reason to possibly cause some confusion among people as to what these acronyms mean.**

Due to the broader use of the MAC acronym, we will use MSC and MAC terms so that we keep coherence between them.

**Lines 71-73: Not all airborne campaigns are of short-duration as your statement suggests. The study reported in Sheridan et al. (2012) had flights 2-3 times per week for 3.25 years!**

Indeed, we will point this out to avoid confusion. Therefore, following the reviewer comment the sentence was modified as follows:

"In North-America, Sheridan et al. (2012) performed flights between 2 and 3 times per week during 3 consecutive years, measuring more than 400 vertical profiles of aerosol properties and trace gases concentrations."

**Line 146: '. . .and the location and ID codes (P1-P7) of the instrumented flights (Table 1).' These are 7 single points on the map. What do they represent? Center of the flight area? Center of profile location? The aircraft covers some horizontal area with its**

**sampling. Possibly better to show shaded areas (or larger dots) for each flight.**

They represent the point of the first measurement showed in the profiles. Following the Reviewer suggestion, we modified the Figure using larger dots to represent the horizontal area covered by the flights. The new Figure is reported below. .

[Figure]

**Line 175: '. . .(calculated for 7th and 16th July 2015). . .'. Why was a trajectory at the start of your first period and at the end of the second period chosen? Were the trajectories consistent throughout each period? If so, why not calculate trajectories in the middle of each period?**

Regarding the dust outbreak period, we chose as beginning of the period the one when the concentration of dust was the highest over the area in order to have a greater picture of the dust outbreak over the area. The trajectory of the REG was calculated for the 16[th] July since during this day small concentrations of dust were also present in the area. Therefore, in order to show both the regional nature of the air masses and the small concentration of dust, we preferred to characterize the whole measurement period. Below we report the trajectories for the 8[th] and 14[th] July, which were included in the supplementary material. The blue trajectory for the 8[th] of July showed that the air mass that originated in the North coast of Africa at 5 km a.g.l. ended over the measurement site at around 1.5 km a.s.l.. For the 14[th] of July, the trajectories at 1.5 and 2.5 km a.g.l. height showed the regional recirculation conditions with the typical zonal winds due to the Azores high pressure, whereas the air mass ending at 750 m a.g.l. showed the local stagnation conditions closer to the surface..

We have added the figures below to the supplementary material.

[Figure]

**Section 2.2.2., first paragraph: Somewhere in this section the sampling inlet needs to be discussed. How far away from the fuselage is the aerosol inlet? How far forward (assuming it is forward) of the leading edge of the wing is the inlet? How far away is the propeller? What is the total flow into this inlet? What is the typical true air speed of the airplane? There is a photograph in Fig. 4 of the airplane which is too small to show any of this information, and dimensions and inlet orientation relative to the wing are not provided in the drawing.**

The aerosol inlet is around 5 cm from de fuselage just where the cockpit changes its shape. It is around 1.5 m forward of the leading edge of the wing and 2.5 meters away from the propeller in the back of the wing.

The inlet flow is set to 10 L/min. The flight characteristics are a speed of 90 kts (160 km/h), a vertical ascent speed of around 500 ft/min with a turning radius of around 900 m. During the descents, the vertical speed and turning radius were slightly larger.

We included this information in the manuscript as follows:

"[…] was designed to be close to isokinetic at a flight speed of 160 km/h thus minimizing the inlet losses. The inlet flow was set to 10 l/min. It was mounted around 5 cm from the fuselage just where the cockpit changes its shape. It was placed around 1.5 m away in front of the leading edge of the wing and 2.5 m away from the propeller in the back of the wing."

**Lines 245-248: Were the data from both ascending and descending profiles used in this paper. It is only stated that for most of the cases, the data from the up and down profiles were similar.**

Both ascending and descending measurements were used in this paper. However, the measurements displayed were those that had the largest data availability. We have added a sentence (reported below) to this paragraph to clarify this point.

"*Whether to present an upward or a downward vertical profile was decided based on which of the two had better data availability and vertical resolution.*"

**Line 249: This statement suggests that only the ascending profiles were used. Please clarify.**

The profiles were performed in a helical way both upwards and downwards, we have clarified this in the text (see our response to previous comment).

"*The method used to perform the vertical profiles consisted of vertical ascensions/descents following helical trajectories (Font et al., 2018), thus allowing the measurement of the aerosol particles properties around a more constrained area*".

**Line 249: 'Helical ascensions' implies that the aircraft was continuously turning. Was the aerosol inlet oriented to account for turning so that aerosols came straight into the inlet? If not, the aspiration efficiency of an inlet not precisely aligned with the flight streamlines should be checked. Was it? Another point. . . unless you have a moveable inlet that can adjust in flight, it is better to stick with one type of profile (ascending, descending, or level flight segments) so that the inlet orientation relative to the streamlines is constant. The pitch of a small aircraft can be different by as much as 10 degrees during ascending and descending profiles, and can also vary with fuel load and air speed. This also causes misalignment of the inlet with the air streamlines going past the inlet and could affect the aspiration efficiency of particles into the inlet. Do you have any calculations or fluid dynamics modeling that suggest this is not important (because you are not discussing it). The fact that aerosol data are similar in 'most' ascending and descending profiles provides circumstantial evidence that this is not a major concern but testing/modeling the effect would provide additional strong evidence.**

The inlet was oriented always in the same direction, i.e. parallel to the airplane, but the turns during the vertical helical motions were made with a radius large enough (around 900 m) to assume that the particles entered straight into the inlet.

Moreover, we noticed only slight differences between the ascending and descending measurements, thus suggesting no major issue related with the fact that the airplane was continuously turning or to differences in the measurement between ascents and descents. Below we attach the ascent and descent profiles of 16[th] July first flight (P6 showed the downward measurement profile of this flight). It can be appreciated that the upward measurement in this profile lacks the measurement of the lower part of the profile between 1. 0 and 1.5 km a.s.l. Nevertheless, both profiles show the same structure, with a peak in the concentration of particles around 1.7 km a.s.l. followed by a decrease in concentration up to around 2.2 km a.s.l. where it increases again and afterwards the concentrations are fairly constant up to 3.5 km a.s.l..

Upwards measurement profile.

[Figure]

Downward measurement profile.

[Figure]

Furthermore, we have modified the first 2 paragraphs of the Subsection 2.2.2. so that we address also the possible measurement errors and losses.

"*Flights were performed with an aircraft Piper PA 34 Seneca (Fig 4) over NE Spain (Fig. 1) from around 0.5 up to 3.5 km a.s.l.. Both the upward and downward flights had approximately the same duration (20-30 min) (Table 1) and the ascent and descent trajectories were performed in the same area when possible. The method used to perform the vertical profiles consisted of vertical ascensions/descents following helical trajectories (Font et al., 2018), thus allowing the measurement of the aerosol particles properties around a more constrained area. The turns during the vertical helical motions were made with a radius large enough (around 900 m) to assume that the particles entered straight into the inlet. The*

*flights were performed at a speed of around 160 km/h with a vertical speed of 152 m/min, slightly larger for the downwards flights. However, some losses may arise due to the perturbation of the flow by the airplane during the sampling, the change in the pitch angle of the aircraft due to the fuel load variation, and ascending vs descending vertical motion. For most of the cases, the up and down profiles were similar, showing therefore the representativeness of the measured profiles and a minimum interference of the aircraft emissions and turbulence. Whether to present an upward or a downward vertical profile was decided based on which of the two better had better data availability and vertical resolution.*

*During the flights, an external volume flow controller (IONER PFC- 6020) was connected to the nephelometer in order to assure a constant sampling flow (5 L/min). Similarly to the in-situ surface measurements, the scattering measurements were performed at RH<40% (GAW, 2016). During the SDE period the vertical profiles of scattering were collected using the Kalman filter available in the AURORA3000 nephelometers, whereas during the REG period vertical profiles this filter was switched off and the raw $\sigma_{Sp}$ and $\sigma_{Bsp}$ coefficients were collected. As shown later, the Kalman filter had the effect of smoothing the measured scattering and hemispheric backscattering coefficients."*

**Lines 251-252: '. . .in order to assure a constant sampling flow (5 Lmin−1).' Was the flow controller a mass flow controller or a volume flow controller?**

It was a volume flow controller. We accordingly added this information to the manuscript as follows:"*an external **volume flow controller** (IONER PFC- 6020)*"

**Lines 257-258: 'Aerosol light-absorption coefficients measurements at seven different wavelengths (370 to 950 nm) were performed with the AE33 aethalometer...'. How were the light absorption coefficients derived for the AE-33 aethalometer (i.e., which correction method was used to derive the aerosol light absorption coefficients from the raw aethalometer data)?**

The AE33 data were corrected online for filter loading effect by the instrument dual-spot correction algorithm (Drinovec et al., 2015) and were presented at ambient standard pressure and temperature. The absorption was derived from the BC concentrations using the MAC ($\lambda$) for the AE33 and the multiple scattering correction factor C=3.15 from Drinovec et al. (2015).

**Line 274: 'The inlet, manufactured by Aerosol d.o.o. (www.aerosol.si), was designed to be close to isokinetic. . .'. At what flow rate was it designed to be close to isokinetic?**

The inlet was designed to be isokinetic at a flight speed of around 160 km/h (airplane speed was 160 km/h) with a flow rate set to 10 lpm.

**Lines 282-283: 'These differences were low considering the spatial variation of the PMx concentrations between P1 flight and MSA station (approximately 10 km).' Up until this point you have not presented any data showing horizontal inhomogeneity of aerosols in the region.**

The reviewer is correct, and no data about the PM homogeneity were shown until now. The last paragraph of section 2.2.2. Airborne measurements were modified to reinforce the results of the PLC (particle loss calculator). According to this tool, losses of large particles in the sampling system were expected. The comparison between the PM1 and PM2.5 concentrations measured at MSA and the lowest point of the flight performed close to MSA evidenced small differences in these size fractions, while large difference was observed for PM10 (~46%). Therefore, confirming particle losses of bigger particles. This fact is important for the discussion of the data collected during the Saharan dust event when large particles are expected to contribute more.

We have rewritten this paragraph in order to avoid any confusion.

*"The inlet (Fig. 4), manufactured by Aerosol d.o.o. (www.aerosol.si), was designed to be close to isokinetic at a flight speed of 160 km/h thus minimizing the inlet losses. The inlet flow was set to 10 l/min. It was mounted around 5 cm from the fuselage just where the cockpit changes its shape. It was placed around 1.5 m away in front of the leading edge of the wing and 2.5 m away from the propeller in the back of the wing. In order to determine aerosol sampling efficiency and particle transport losses we used the Particle Losses Calculator(PLC) software tool (von der Weiden et al., 2009). The results (not shown) indicated that the losses at the inlet were minimal for PM2.5 and that the losses inside the sampling system were large for dust particles larger than around 4-5μm. To further confirm the PLC results, we compared the PMx measurements at MSA station with the PMx aircraft measurements performed at the same altitude of MSA during the closest flight to MSA at a distance of 10 km. This flight (P1; 7th July 2015; Fig.1) was performed during a Saharan dust outbreak thus the presence of dust particles in the atmosphere was significant. Table S1 shows that differences were low, around 6% for PM1 and 12% for PM2.5 with the aircraft underestimating the measurements at MSA. However, the PM10 aircraft measurements were around 47% lower compared to the PM10 measurements performed at MSA thus confirming the PLC results. Therefore, the inlet losses for particles other than dust were minimal. In fact, Table S1 shows that for scattering and absorption measurements (which were performed in the PM10 fraction at MSA) the differences were <9% and <16%, respectively because of the high scattering efficiency of fine particles compared to coarse particles (e.g. Malm and Hand, 2007) and because the absorbing fraction is mostly contained in the fine aerosol particle mode."*

**Lines 307-309: 'For the vertical profiles reported here, the AAE was calculated using the seven AE33 wavelengths when all the seven absorption measurements were positive. For some profiles, the lower wavelengths (370 to 590, 660, or 880 nm) were used for the calculation of AAE.' Why was the method not kept consistent for all profiles?**

For some profiles, the absorption measurements at longer wavelengths were slightly negative due to the very low particles absorption properties measured at high altitudes at these wavelengths. Since AAE can only be derived when absorption coefficients are positive, different wavelength ranges were chosen for the calculation of the AAE. In this study, the AAE was used to identify different atmospheric conditions (dust versus regional episodes) as well as to distinguish different layers within the same flight. Consequently, we think that the wavelength pair used is not that relevant since the focus was kept on the variability of

AAE rather than on its absolute vale. Thus, the effect of using different wavelength pairs on AAE is out of the scope of this manuscript. For all the profiles, except the flight P3, the first 4 wavelengths were used. For the sake of clarity, the wavelength pair used in each flight has been included in the figures, as well as in the main text.

*"For the vertical profiles reported here, the AAE was calculated using the AE33 wavelengths for which the absorption measurements were positive along the profile. For most profiles, except for P3, which had all seven wavelengths available, the AAE was calculated from 370 to 590 nm."*

**Lines 313-314: 'The SSA was obtained by extrapolating the total scattering at the Aethalometer wavelengths using the SAE.' Supposed to be 'SSAAE' instead of SSA? How confident are you in extrapolating the scattering, which is measured in the visible range, into the near UV and near IR?**

Indeed, the SSAAE can be determined from the SSA calculated at different wavelengths. However, here we meant SSA which was extrapolated at the seven AE33 wavelengths using the experimental SAE. The part of the text related to this point has been changed as follows:

*"The Single Scattering Albedo Ångström Exponent (SSAAE) was calculated by fitting the SSA retrieved at the same Aethalometer wavelengths used to calculate the AAE. For this, the SSA was obtained by extrapolating the total scattering to the AE33 wavelengths using the calculated SAE.".*

To extrapolate the scattering at UV and near-IR wavelengths we assumed a linear relationship for the scattering at the different wavelengths in a log-log space. The feasibility of this approach has been demonstrated in previous papers (e.g. Collaud Coen et al., 2004; Ealo et al., 2016) where the corresponding SSAAE was successfully used for the detection of desert dust in the atmosphere. Given that the extrapolation of the scattering at UV and near-IR wavelengths was used in the present manuscript for the calculation of the SSAAE (and dust detection), we assume that the methodology we applied is valid.

**Lines 323-324: '. . .presented in Fig. 5. . .' Some text in Fig. 5 barely readable. Font should be a little larger if possible.**

We have increase the font of the text in Fig. 5 to make it more readable. We include below one of the figures included in Fig. 5 as an example of the increased font of the text within Fig. 5 subfigures. Also, we would like to point out that in the conversion from the discussion manuscript to a 2-column text, there is space to increase the size of the figure.

[Figure]

**Lines 387-388: 'The REG episode resulted in the development of aerosol layers at high altitude, as also observed with the ceilometer at MSA (Figs. 6c,d). How do you know that the features in these retrievals are not from light or subvisible cirrus rather than aerosols?**

Indeed, ceilometers are not able to distinguish among the different possible scatterers in the atmosphere mostly because these instruments work at only one wavelength. Nevertheless, the intensity of the backscatter signal from the aerosol strata during the regional episodes were not as strong as to indicate the presence of clouds. Moreover, we presented ceilometer data below 6 km thus probably allowing to exclude the presence of high-altitude cirrus clouds. Furthermore, the AERONET cloud-screened level 2 data from a sun-photometer installed at the MSA station were available during the airborne measurements thus further excluding the presence of clouds.

[Figure]

**Lines 395-397: 'The scattering (Fig. 5e) and absorption coefficients 395 (Fig. 5f) at both sites were above the average values (Table 2) presented by Pandolfi et al. (2014a) and Ealo et al. (2016) for REG episodes, confirming the accumulation of pollutants since the 10th July onward.' I would not phrase this statement in this way. The fact that the scattering and absorption values here were higher than those reported in other studies**

**from different years DOES NOT confirm the accumulation of pollutants over this period in your study. These are separate and unrelated events. Pollutants were indeed accumulating during this study, but it is not because they were higher than previously-measured average values. This point should be rephrased in the manuscript.**

We thank the Reviewer for the suggestion. We have rephrased the sentence as follows:

"*The scattering (Fig. 5e) and absorption coefficients (Fig. 5f) at both sites were above the average values (Table 2) presented by Pandolfi et al. (2014a) and Ealo et al. (2016) for REG episodes, due to the strong accumulation of pollutants which took place since 10th July onward for this particular REG episode*".

**Lines 426-427: 'Figure 7 shows the three vertical profiles during the SDE period...'. Same question as earlier. . . are these data from ascent profiles, descent profiles, an average of the two profiles, or what? If you are averaging two profiles, you can be averaging out real features in both profiles. Also, please state what the shaded envelopes represent for the intensive profile properties in Figs. 7 and 8.**

We agree with the Reviewer. The profiles presented here were either ascendant or descendent, and never an average of both. We clarified this point in the manuscript in Table 1.

The shadowed envelopes of the intensive properties represent the calculated standard error of the variables. In order to clarify this point, the Figure caption was accordingly modified.

**Lines 445-447: Certainly there was dust, but can you rule out BrC or OA also contributing to the higher AAE's?**

Indeed, BrC (and OM) can cause an increase of AAE because of its high UV-VIS absorption properties However, given that during the REG episode a similar increase of AAE was not observed, we assumed that the AAE increase during the dust attributed episode was mostly due to the presence of dust particles rather than to BrC. Moreover, the filter analyses at MSY (Fig. 5b) confirmed the increase of dust particles during the first episode versus the REG episode, whereas EC and OM remained fairly similar.

**Line 458: 'As already noted, the observed AAE increase at these altitudes was due to the UV absorption from dust particles.' Still not sure how you have ruled out BrC/OA co-existing with the dust. It happens frequently with Asian dust where the pollution aerosols are transported along with the dust because the air mass travels over both dust and pollution sources and picks up both types of aerosols.**

Indeed, African air masses can be enriched in anthropogenic pollutants during the transport from North African deserts to our region. However, as shown in Rodriguez et al. (2011), major sources of pollutants mixed with dust are mostly industrial emissions occurring in Northern Algeria, Eastern Algeria, Tunisia and the Atlantic coast of Morocco that appear as the most important source of the nitrate and a fraction of sulphate. Unlike Asian dust, African

air masses reach the Iberian Peninsula without travelling above regions with strong biomass burning or coal combustion emissions, both being potentially important sources of BrC. In addition, the in-situ surface filter analysis measurements performed at MSY station confirmed the predominance of dust.

**Bibliography.**

1. Sheridan, P. J., Andrews, E., Ogren, J. A., Tackett, J. L. & Winker, D. M. Vertical profiles of aerosol optical properties over central Illinois and comparison with surface and satellite measurements. *Atmos. Chem. Phys.* **12**, 11695–11721 (2012).

2. Collaud Coen, M., Weingartner, E., Schaub, D., Hueglin, C., Corrigan, C., Henning, S., Schwikowski, M., and Baltensperger, U.: Saharan dust events at the Jungfraujoch: detection by wavelength dependence of the single scattering albedo and first climatology analysis, *Atmos. Chem. Phys*, **4**, 2465–2480 (2004).

3. Drinovec, L. *et al.* The 'dual-spot' Aethalometer: An improved measurement of aerosol black carbon with real-time loading compensation. *Atmos. Meas. Tech.* **8**, 1965–1979 (2015).

4. Rodríguez, S., Alastuey, A., Alonso-Pérez, S., Querol, X., Cuevas, E., Abreu-Afonso, J., Viana, M., Pérez, N., Pandolfi, M., and de la Rosa, J.: Transport of desert dust mixed with North African industrial pollutants in the subtropical Saharan Air Layer, *Atmos. Chem. Phys*., **11**, 6663–6685, https://doi.org/10.5194/acp-11-6663-2011, 2011.

5. Ealo, M., Alastuey, A., Ripoll, A., Pérez, N., Minguillón, M. C., Querol, X., and Pandolfi, M.: Detection of Saharan dust and biomass burning events using near-real-time intensive aerosol optical properties in the north-western Mediterranean, *Atmos. Chem. Phys*., **16**, 12567–12586, https://doi.org/10.5194/acp-16-12567-2016, 2016.

6. Sheridan, P. J., Andrews, E., Ogren, J. A., Tackett, J. L. & Winker, D. M. Vertical profiles of aerosol optical properties over central Illinois and comparison with surface and satellite measurements. *Atmos. Chem. Phys.* **12**, 11695–11721 (2012).

Atmos. Chem. Phys. Discuss.,
https://doi.org/10.5194/acp-2020-837-RC2, 2020

[Figure]

Summary: The manuscript summarizes findings from an aircraft campaign over the western Mediterranean Basin in conjunction with in-situ aerosol measurements at two monitoring sites- MSY and MSA. The analysis explores two main episodes during the campaign, designated as (1) regional pollution episodes and (2) Saharan dust events. The paper synthesizes measured aerosol optical properties- including extensive and intensive properties- from the campaign and highlights the differences in meteorology and aerosol populations during the two types of episodes. While no new methods are presented, it is clear that care has been taken with the data and the analysis. The

paper is very well written and the analysis is clear.

General Comments:

-Excellent background section!

-My main qualm with the paper is that Figure 7 and 8 are quite difficult to read. It is as if the ratio of the figures is off; they should be slightly wider. I realize the challenges associated with presenting this much information at once- and I see the logic behind how these figures are organized. If the individual plots can be widened slightly or maybe if the legends could be moved to inside the plots to allow them to get bigger, that might help the reader. But the paper could be published with the figures as is if necessary.

Technical Comments:

Line 46: Perhaps add a reference to the NOAA Federated Aerosol Network here as well?

Line 65: Suggest changing to '. . .one or two a day' or '. . .once or twice a day'

Line 111-112 : Grammatically, 'reach' should be 'reaches' and 'create' should be 'creates'

Line 175: Suggest changing 'hysplit' to 'HYSPLIT' since their website capitalizes it

Figure 2: It would be nice to add the units of measurement to the figure descriptions in the caption, as they are difficult to read on the plots

Line 216: Please specify if aethalometer data were further corrected or if the manufacturer 2-spot correction was the correction used

Line 216: Were any corrections applied to the MAAP data?

Line 265: Extra ')' on this line can be deleted

Lines 307-309: It sounds like AAE was calculated using different wavelength pairs for

different legs. What is the effect of this on your results? Why not use a wavelength pair available in all flight leg data records so you have consistency?

Line 323: Add a space after 'concentrations,' before 'PM1/10'

Line 324: Add a space after 'sigma[ap]'

Line 358: It seems like 'typically register' should be past tense 'typically registered'

Line 386: should say 'also took place'

Line 536: How was the PBL estimated?

Lines 559-560: 'associated to' should be 'associated with'

Line 595: remove 'was'

Line 601: 'wildfire-related' instead of 'wildfires related'

Line 603-605: This sentence is difficult to read; consider rewording.
* * *
On behalf of all the authors, we thank the Reviewer for the positive comments and careful review, which helped improve the manuscript.

Hereafter we reply to the Reviewer's comments. Any minor comment as typo or writing corrections will be directly corrected in the manuscript.

**General comments**

- We thank the Reviewer for the positive comment regarding the work done by our research group and regarding the scientific significance of the present manuscript.

**-My main qualm with the paper is that Figure 7 and 8 are quite difficult to read. It is as if the ratio of the figures is off; they should be slightly wider. I realize the challenges associated with presenting this much information at once- and I see the logic behind how these figures are organized. If the individual plots can be widened slightly or maybe if the legends could be moved to inside the plots to allow them to get bigger, that might help the reader. But the paper could be published with the figures as is if necessary.**

We have taken into account the suggestion and modified the figures so that these are now a bit wider. Below an example of the new format for P1. Also, we would like to point out that in the conversion from the discussion manuscript to a 2-column text, there will be more space to increase the size of the figure.

[Figure]

**Technical comments**

**Line 216: Please specify if aethalometer data were further corrected or if the manufacturer 2-spot correction was the correction used**

The AE33 data were corrected online for filter loading effect by the instrument dual-spot correction algorithm (Drinovec et al., 2015) and were presented at ambient standard pressure and temperature. The absorption was derived from the BC concentrations using the MAC ($\lambda$) for the AE33 and the multiple scattering correction factor C=3.15 from Drinovec et al. (2015).

**Line 216: Were any corrections applied to the MAAP data?**

The MAAP absorption reported in this study was measured at 637 nm, whereas the nominal MAAP wavelength is 670 nm. As shown in Muller et al. (2011) this difference in the wavelength can be taken into account by multiplying the absorption data provided by the MAAP by 1.05, as we did in this work. In order to clarify this point, the following sentence was added to the second paragraph on Section 2.2.1.

"MAAP data in this work were reported at 637 nm by multiplying the MAAP absorption data by a factor of 1.05 as suggested by Muller et al. (2011)".

**Lines 307-309: It sounds like AAE was calculated using different wavelength pairs for different legs. What is the effect of this on your results? Why not use a wavelength pair available in all flight leg data records so you have consistency?**

For some profiles, the absorption measurements at longer wavelengths were slightly negative due to the very low particles absorption properties measured at high altitudes at these wavelengths. Since AAE can only be derived when absorption coefficients are positive, different wavelength ranges were chosen for the calculation of the AAE. In this study, the AAE was used to identify different atmospheric conditions (dust versus regional episodes) as well as to distinguish different layers within the same flight. Consequently, we think that the wavelength pair used is not that relevant since the focus was kept on the variability of AAE rather than on its absolute vale. Thus, the effect of using different wavelength pairs on AAE is out of the scope of this manuscript. For all the profiles, except the flight P3, the first 4 wavelengths were used. For the sake of clarity, the wavelength pair used in each flight has been included in the figures, as well as in the main text.

*"For the vertical profiles reported here, the AAE was calculated using the AE33 wavelengths for which the absorption measurements were positive along the profile. For most profiles, except for P3, which had all seven wavelengths available, the AAE was calculated from 370 to 590 nm."*

**Line 536: How was the PBL estimated?**

It was inferred from both the ceilometer profiles and the potential temperature and relative humidity measurements obtained by the aircraft during the vertical profiles.

*"The lower AAE measured at MSA and MSY stations compared to the AAE in the free troposphere measured during P1, P2 and P3 was due to the increased relative importance of BC particles close to the ground and within the PBL (estimated from the observation of the pollutant concentrations, the ceilometer profiles and the meteorological conditions: potential temperature and relative humidity in Fig. S3)"*

**List of all relevant changes made in the manuscript:**

- ✓ Updated the figures 1, 5, 7 and 8 on the manuscript so that they are easier to interpret.
- ✓ Clarified the aircraft sampling and the potential measurements errors related with it.

[revised manuscript text omitted]

* * *
[35]removed: measurements with the aircraft

[36]removed: seven

[37]removed: when all the seven

[38]removed: . For some profiles, the lower wavelengths (

[39]removed: , 660, or 880 nm) were used for the calculation of AAE

325     d. The Single Scattering Albedo (SSA) reported here was calculated as the ratio between the scattering and the extinction coefficients at 525 nm. SSA indicates the potential of aerosols for cooling or warming the atmosphere.

    e. The Single Scattering Albedo Ångström Exponent (SSAAE) was [..[40] ]calculated by fitting the SSA retrieved at the same Aethalometer wavelengths used to calculate the AAE. [..[41] ]For this, the SSA was obtained by extrapolating the total scattering [..[42] ]to the AE33 wavelengths using the calculated SAE. SSAAE is a good indicator for the the presence
330     of coarse particles (e.g. dust) when values are <0 (Coen et al., 2004; Ealo et al., 2016).

    f. The Mass Scattering Cross-section (MSC) is the ratio between the scattering and the PM concentration. It represents the scattering efficiency of the collected particles per unit of mass.

    g. The Mass Absorption [..[43] ]Cross-section (MAC) is the ratio between the light absorption and the PM concentration. It represents the absorption efficiency of the collected particles per unit of mass.

335 # 4 Results

**4.1 MSA and MSY in situ measurements**

The time evolution of $PM_1$, $PM_{10}$ concentrations, $PM_{1/10}$ ratios, BC, PM components (as measured by ACSM at MSY), $\sigma_{ap}$, $\sigma_{sp}$, SAE, AAE, g and SSA measured at MSY and MSA during the first three weeks of July 2015 is presented in Fig. 5 together with the concentrations of major species ($NO_3^-$, $SO_4^{2-}$, $NH_4^+$, EC, OM (with an OM/OC ratio of 2.1), mineral matter (MM;
340 calculated as the sum of typical mineral oxides) and sea salt (SS; Na + Cl)) from offline analysis of 24h filters collected at MSY and MSA during the days of the instrumented flights. Table 2 shows the mean values of surface $PM_1$, $PM_{10}$, $PM_{1/10}$, $\sigma_{sp\ 525nm}$, $\sigma_{ap\ 637nm}$, SAE, AAE, SSA and g measured at MSY and MSA stations on 7th-8th and 14th-16th July compared to the mean values typically measured during SDE and REG pollution episodes as reported by Pandolfi et al. (2014b) and Ealo et al. (2016).

345     As shown in Fig. 5 an accumulation of pollutants took place from 4th to 8th July, as evidenced by the gradual increase of concentrations of $SO_4^{2-}$, BC, OA, PM, total scattering and absorption coefficients. This accumulation was favored by the regional stagnation and vertical recirculation of the air masses. Moreover, a Saharan dust outbreak caused a progressive increase of $PM_{10}$ at both stations, as well as a simultaneous $PM_{1/10}$ ratio decrease (Fig. 5c,d). The dust event had a larger impact at MSA where the $PM_{10}$ levels increased sharply and were higher compared to MSY. As reported in Table 2 and in Fig. 5, starting from
350 6th July, the $PM_{10}$ concentrations were higher compared to average $PM_{10}$ usually measured during Saharan dust outbreaks at both stations. Then, the levels of pollutants, as well as the total scattering and absorption coefficients, decreased on 9th July due to the venting of the basin by an Atlantic North West (ANW) advection (not shown) that cleansed the northern area of the
* * *
[40] removed: obtained as fit of the SSA calculated

[41] removed: The

[42] removed: at the Aethalometer

[43] removed: Efficiency (MAE

[revised manuscript text omitted]